# Automatic Unsupervised Outlier Model Selection

**Yue Zhao**
Carnegie Mellon University
`zhaoy@cmu.edu`

**Ryan A. Rossi**
Adobe Research
`ryrossi@adobe.com`

**Leman Akoglu**
Carnegie Mellon University
`lakoglu@andrew.cmu.edu`

## Abstract

Given an unsupervised outlier detection task on a new dataset, how can we automatically select a good outlier detection algorithm and its hyperparameter(s) (collectively called a model)? In this work, we tackle the *unsupervised outlier model selection* (UOMS) problem, and propose METAOD, a principled, data-driven approach to UOMS based on meta-learning. The UOMS problem is notoriously challenging, as compared to model selection for classification and clustering, since ($i$) model evaluation is infeasible due to the lack of hold-out data with labels, and ($ii$) model comparison is infeasible due to the lack of a universal objective function. METAOD capitalizes on the performances of a large body of detection models on historical outlier detection benchmark datasets, and carries over this prior experience to automatically select an effective model to be employed on a new dataset *without any labels, model evaluations or model comparisons*. To capture task similarity within our meta-learning framework, we introduce specialized meta-features that quantify outlying characteristics of a dataset. Extensive experiments show that *selecting* a model by METAOD significantly outperforms no model selection (e.g. always using the same popular model or the ensemble of many) as well as other meta-learning techniques that we tailored for UOMS. Moreover upon (meta-)training, METAOD is extremely efficient at test time; selecting from a large pool of 300+ models takes less than 1 second for a new task. We open-source[1] METAOD and our meta-learning database for practical use and to foster further research on the UOMS problem.

## 1 Introduction

The lack of a universal learning model that performs well on *all* problem instances is well recognized [53]. Therefore, effort has been directed toward building a toolbox of various models and algorithms, which has given rise to the problem of algorithm selection and hyperparameter tuning (i.e., model selection). The same problem applies to outlier detection (OD); a long list of detectors has been developed in the last decades [2], with no universal "winners" [8].

In supervised learning, model selection can be done via performance evaluation of each trained model on labeled hold-out data. In contrast, *unsupervised* OD does not have access to any labels, nor is there a universal objective function that could guide model selection (cf. clustering where a loss function enables model comparison). Unsupervised model selection for OD is challenging exactly because both model evaluation and comparison are *not* feasible—which renders any trial-and-error techniques like grid search or iterative strategies like Bayesian hyperparameter optimization [57] inapplicable. Consequently, there has been no principled work on unsupervised outlier model selection—rather, the choice of a model for a new task (or dataset) remains "a black art". A typical approach is to use popular OD algorithms, like LOF [6] and iForest [31] (often with default hyperparameters) which are shown to be competitive on average on many benchmark datasets. However, as noted earlier, none of

---

[1]Code available at URL: `https://github.com/yzhao062/UOMS`

35th Conference on Neural Information Processing Systems (NeurIPS 2021).

these methods can universally outperform others on all tasks [8]. We argue that model selection is exactly how one can "break the performance ceiling" for OD.

In this work, we tackle the unsupervised outlier model selection (UOMS) problem systematically. To that end, we introduce (to the best of our knowledge) the first UOMS approach that selects an effective model to be employed on a new detection task *without any model evaluation (using labels) or model comparison (via loss criteria)*. Our proposed method, called METAOD, is based on meta-learning, and stands on the prior performances of a large collection of existing detection models on an extensive corpora of historical outlier detection benchmark datasets. In a nutshell, the idea is to estimate a candidate model's performance on the new task (with no labels) based on its prior performance on *similar* historical tasks. We remark that METAOD is strictly a model selection technique – that picks one model (a detector and its associated hyperparameter(s)) from a pool of (existing) candidate models – and *not* yet-another outlier detection algorithm itself.

In leveraging meta-learning, we establish a connection between the UOMS problem and the cold-start problem in collaborative filtering (CF), where the new task in UOMS is akin to a new user in CF (with no available evaluations, hence cold-start) and the model space is analogous to the item-set. Differently, OD necessitates the identification of a single best model (i.e., top-1 rank), whereas CF typically operates in a top-$k$ setting. In CF, future recommendations can be improved based on user feedback which is not applicable to OD. Moreover, METAOD requires the effective learning of task similarities based on characteristic dataset features (namely, meta-features) that capture the outlying properties within a dataset, whereas user features (location, age, etc.) in CF may be readily available.

In summary, the key contributions of this work include the following:

- **First Approach to Unsupervised Outlier Model Selection**: We propose METAOD, (to our knowledge) the first effort on *unsupervised model selection* for OD tasks. Notably, given a new dataset (i.e., at test time), it does not rely on any ground-truth labels for model evaluation or any loss or heuristic criterion for model comparison. METAOD stands on meta-learning in principle, and historical collections of outlier models and benchmark datasets in practice.
- **Problem Formulation:** We establish a *correspondence between UOMS and CF under cold-start*, where the new task "better likes" a model that performs better on similar historical tasks.
- **Specialized Meta-features:** We design novel meta-features to capture the outlying characteristics in a dataset toward effectively quantifying task similarity specifically among OD tasks.
- **Effectiveness and Efficiency**: Through extensive experiments on two benchmark testbeds that we have constructed, we show that *selecting* a model by METAOD for each given task significantly outperforms always using a popular model like iForest, as well as other possible meta-learning approaches that we tailored for UOMS. Moreover, METAOD incurs negligible run-time overhead (<1 second) at test time.
- **Open-source Platform**: We open-source[1] METAOD and our meta-learning database for the community to use it for UOMS in practice, and to extend it with new datasets and models. We expect the growth of the database would make meta-learning based approaches, like METAOD, more powerful and also help foster further research on this new direction to an important problem.

## 2   Related Work

### 2.1   Model Selection for Outlier Detection (OD)

Most outlier mining work have focused on developing new, better methods for *detection* on different types of data [2]. In comparison, there are only a few work on the outlier model *selection* problem –which detector and hyperparameter(s) to use on a new task– all of which require some labeled data. Recent work include AutoOD [29] that focuses on automatic neural architecture search, however, it is limited to deep autoencoder based detection, and more importantly it relies on hold-out labeled data for evaluation. Similarly, PyODDS [30] and TODS [25] both require ground truth labels.

To our knowledge, there is *no* existing work on the general *unsupervised* outlier model selection (UOMS) problem, which is considerably more challenging, as (1) model evaluation is infeasible due to the lack of any ground truth labels, and (2) model comparison of different heterogeneous detectors is infeasible due to the lack of a universal OD loss function.

We note that *some* OD methods do have a loss function; e.g. auto-encoders [9, 63] aim to minimize reconstruction error for modeling the inliers, and one-class classification (OCSVM [38], SVDD [46], etc.) aims to maximize margin to the origin or minimize the radius of a data-enclosing hyperball. There is also work on model selection for one-class models [7, 54], however, those are limited to this specific family of methods and do not apply in the general case. Our proposed METAOD is not limited to one specific model family, but can select among any (heterogeneous) set of detectors.

## 2.2 Model Selection in ML, AutoML, Meta-Learning

Model selection refers to the process of algorithm selection and/or hyperparameter optimization (HO). With the advent of complex (e.g. deep) models, HO in high dimensions has become impractical to be human-powered [59]. As such, automating ML pipelines has seen a surge of attention [18]. Meta-learning has been a key contributor to the AutoML effort [41, 49, 58].

**Supervised Model Selection:** Most existing work focus on the supervised setting, and use hold-out data with labels. Randomized [3], bandit-based [27], and Bayesian optimization (BO) techniques [40] are various state-of-the-art (SOTA) approaches to HO. Specifically sequential model-based BO [19, 22] evaluates hold-out performance at various initial hyperparameter configurations (HC), where a (smooth) surrogate function is fit to the resulting ⟨HC, performance⟩ pairs, which is then used to strategically query other HCs, e.g., via hyper-gradient based search [15]. Meta-learning has also been employed [13, 52], e.g., to find promising initialization for (i.e. warm-starting) BO [14, 51].

Note that *all of these approaches rely on multiple model evaluations* (i.e., performance queries) for various HCs, and hence cannot be applied to the *unsupervised* outlier model selection problem.

**Unsupervised Model Selection:** Unsupervised ML tasks (e.g., clustering) poses additional challenges for model selection [12, 47]. Nonetheless, those exhibit *established objective criteria* that enable model comparison, *unlike* OD. For example, BO methods still apply where the surrogate can be trained on ⟨HC, objective value⟩ pairs, for which meta-learning can provide favorable priors.

Task-independent meta-learning [1], that simply identifies the globally best model on historical tasks, applies to the unsupervised setting and hence OD. This can be refined by identifying the best model on not all, but *similar* tasks, where task similarity is measured in the meta-feature space via clustering [23] or nearest neighbors [33]. This type of similarity-based recommendations points to a connection between algorithm selection and collaborative filtering (CF), first recognized by Stern *et al.* [44]. The most related to UOMS is CF under cold start, where evaluations are not-available (in our case, infeasible) for a new user (in our case, task). There have been a number of work using meta-learning for the cold-start recommendation problem [4, 26, 50], and vice versa, using CF solutions for ML algorithm selection [32, 56]. We tailor these to UOMS and compare to METAOD in the experiments.

# 3 Unsupervised Outlier Model Selection via Meta-learning

## 3.1 Problem Statement

We consider the model selection problem for unsupervised outlier detection, which we refer to as UOMS (unsupervised outlier model selection) hereafter. Given a new dataset, *without any labels*, the problem is to select both ($i$) a detector/algorithm and ($ii$) its associated hyperparameter(s) (HP). The former is a discrete choice, given the finite set of existing detection algorithms. The latter is continuous, and hence induces infinitely many candidate models.

Under certain assumptions, such as performance changing smoothly in the HP space, a HP configuration can be selected iteratively based on evaluations on several other carefully-chosen configurations. Importantly however, OD is not amenable for such iterative search over models—evaluations are *not possible* due to the lack of labels and absence of a universal objective criterion. The selection of a model, therefore, is to be done without building or evaluating any model on the new dataset. Given this constraint, we discretize the HP space for each candidate detector to make the search space tractable, which induces a finite pool of models denoted $\mathcal{M} = \{M_1, \ldots, M_m\}$. Each model $M \in \mathcal{M}$ can be seen as a {detector, configuration} pair, where the configuration depicts a specific set of values for the detector's HP(s). (See Appendix A for details.) Then, the UOMS problem is stated as follows:

**Problem 1 (Unsupervised Outlier Model Selection (UOMS))** Given *a new input dataset (i.e., detection task)* $\mathbf{D}_{test} = (\mathbf{X}_{test})$ *without any labels,* Select *a model* $M \in \mathcal{M}$ *to employ on* $\mathbf{X}_{test}$.

## 3.2 Proposed METAOD

In this work we consider the UOMS problem and propose a meta-learning based solution, leveraging past experience on historical detection tasks. As such, our METAOD relies on

- a collection of historical outlier detection datasets $\mathcal{D}_{\text{train}} = \{\mathbf{D}_1, \ldots, \mathbf{D}_n\}$, namely, a meta-train database with ground truth labels, i.e., $\{\mathbf{D}_i = (\mathbf{X}_i, \mathbf{y}_i)\}_{i=1}^n$, and
- the historical performances of the pool of candidate models, $\mathcal{M}$, on the meta-train datasets. We denote by $\mathbf{P} \in \mathbb{R}^{n \times m}$ the performance matrix, where $\mathbf{P}_{ij}$ corresponds to the $j$-th model $M_j$'s performance[2] on the $i$-th meta-train dataset $\mathbf{D}_i$.

Note that model performance can be evaluated on the historical meta-train datasets as they contain ground truth labels, which however is not the case for any newcoming task at test time.

Our METAOD consists of two-phases: **offline (meta-)training** of the meta-learner on $\mathcal{D}_{\text{train}}$, and **online prediction** that enables unsupervised model selection at test time for $\mathbf{D}_{\text{test}}$. Arguably, the running time of the offline phase is not critical. In contrast, model selection for a newcoming task should incur small run-time overhead, as it precedes the actual building of the selected OD model. Fig. 1 summarizes the process and the major components of METAOD, where we highlight the components transferred from offline (meta-learning) to online stage (model selection) in blue. We also provide the detailed steps of METAOD in pseudo-code, for both meta-training (offline) and model selection (online), in Appendix D Algo. 1.

### 3.2.1 (Meta-)Training (Offline)

In principle, meta-learning carries over prior experience on a set of historical tasks to "do better" on a new task. Such improvement can be unlocked only if the new task *resembles* and thus can build on *at least some* of the historical tasks (such as learning ice-skating given prior experience with roller-blading), rather than representing completely unrelated phenomena. This entails defining an effective way to capture task similarity between an input task and the historical tasks at hand.

In machine learning, similarity between meta-train and test datasets are quantified through characteristic features of a dataset, also known as *meta-features*. Those typically capture statistical properties of the data distributions. (See survey [49] for various types of meta-features.)

To capture **prior experience**, METAOD first constructs the performance matrix $\mathbf{P}$ by running/building and evaluating all the $m$ models in our defined model space $\mathcal{M}$ on all the $n$ meta-train datasets $\mathcal{D}_{\text{train}}$.[3] To capture **task similarity**, it then extracts a set of $d$ meta-features from each meta-train dataset, denoted by $\mathbf{M} = \psi(\{\mathbf{X}_1, \ldots, \mathbf{X}_n\}) \in \mathbb{R}^{n \times d}$ where $\psi(\cdot)$ depicts the feature extraction module. We defer the details on the meta-feature specifics to §3.3.

At this stage, it is easy to recognize the connection between the UOMS and the collaborative filtering (CF) under cold start problems. Simply put, meta-train datasets are akin to existing users in CF that have prior evaluations on a set of models that are akin to the item-set in CF. The test task is akin to a newcoming user with no prior evaluations (and in our case, no possible future evaluation either), which however exhibits some pre-defined features.

Capitalizing on this connection, we take a matrix factorization based approach where $\mathbf{P}$ is approximated by the dot product of what-we-call dataset matrix $\mathbf{U} \in \mathbb{R}^{n \times k}$ and model matrix $\mathbf{V} \in \mathbb{R}^{m \times k}$. The intent is to capture the inherent dataset-to-model affinity via the dot product similarity in the $k$-dimensional latent space, such that $\mathbf{P}_{ij} \approx \mathbf{U}_i \mathbf{V}_j^T$ where matrix subscript denotes the row.

What loss criterion is suitable for the factorization? In CF the typical goal is top-$k$ item recommendation. In METAOD, we aim to select the model with the best performance on a task which demands top-1 optimization. Therefore, we discard least squares and instead optimize the rank-based (row- or

---

[2]Area under the precision-recall curve (Average Precision or AP); can be substituted with any other measure.
[3]Note that this step takes considerable compute-time, which however amortizes to "do better" for future tasks. To this effect, we open-source our *trained* meta-learner to be readily deployed.

dataset-wise) discounted cumulative gain (DCG) [21],

$$\max_{\mathbf{U},\mathbf{V}} \ \sum_{i=1}^{n} \mathrm{DCG}_i(\mathbf{P}_i, \mathbf{U}_i\mathbf{V}^T) \ . \tag{1}$$

The factorization is solved via alternating optimization, where initialization plays an important role for such non-convex problems. We find that initializing $\mathbf{U}$, denoted $\mathbf{U}^{(0)}$, based on meta-features facilitates stable training, potentially by hinting at inherent similarities among datasets as compared to random initialization. Specifically, an embedding function $\phi(\cdot)$ is used to set $\mathbf{U}^{(0)} := \phi(\mathbf{M})$ for $\phi : \mathbb{R}^d \mapsto \mathbb{R}^k$, $k < d$. Details on objective criteria and optimization are deferred to §3.4.

By construction, matrix factorization is transductive. On the other hand, we would need $\mathbf{U}_{\text{test}}$ to be able to estimate performances of the model set $\mathcal{M}$ on a new dataset $\mathbf{X}_{\text{test}}$. To this end, one can learn an (inductive) multi-output regression model that maps the meta-features onto the latent features. We simplify by learning a regression function $f : \mathbb{R}^k \mapsto \mathbb{R}^k$ that maps the (lower dimensional) embedding features $\phi(\mathbf{M})$ (which are also used to initialize $\mathbf{U}$) onto the final optimized $\mathbf{U}$. Note that this requires an inductive embedding function $\phi(\cdot)$ to be applicable to newcoming datasets. In implementation, we use PCA for $\phi(\cdot)$ and a random forest regressor for $f(\cdot)$ although METAOD is flexible to accommodate any others provided they are inductive.

**Remark:** METAOD improves over the existing methods that use CF in machine learning model selection (see §2.2) in two aspects. First, METAOD builds specialized landmarker features tailored for capturing outlying characteristics of a dataset, while the existing ML model selection mainly uses generic statistical features (see §3.3). Second, METAOD uses a customized (backpropagatable/smooth) rank-based loss in CF for more effective top-1 optimization (see §3.4), while existing approaches mainly leverage mean squared loss (MSE).

### 3.2.2 Prediction for Unsupervised Model Selection (Online)

Meta-training stage yields the estimated functions $\psi(\cdot)$, $\phi(\cdot)$, and $f(\cdot)$ as well as the model matrix $\mathbf{V} \in \mathbb{R}^{m \times k}$, which we save for test time (See Fig. 1). Given a new dataset $\mathbf{X}_{\text{test}}$ for OD, METAOD first computes the corresponding meta-features as $\mathbf{M}_{\text{test}} := \psi(\mathbf{X}_{\text{test}}) \in \mathbb{R}^d$. Those are then embedded via $\phi(\mathbf{M}_{\text{test}}) \in \mathbb{R}^k$, which are regressed to obtain the latent features, i.e., $\mathbf{U}_{\text{test}} := f(\phi(\mathbf{M}_{\text{test}})) \in \mathbb{R}^k$. Model set performances are predicted as $\mathbf{P}_{\text{test}} := \mathbf{U}_{\text{test}}\mathbf{V}^T \in \mathbb{R}^m$. Finally, the model with the largest predicted performance is outputted as the selected model, that is,

$$\arg\max_{j} \ \langle \, f(\phi(\psi(\mathbf{X}_{\text{test}}))), \mathbf{V}_j \, \rangle \ . \tag{2}$$

**Remark:** Notice that model selection by Eq. (2) for a newcoming dataset is solely based on its meta-features and other pre-trained components from meta-learning. It does not rely on ground-truth labels or any OD model evaluations, therefore, METAOD provides *unsupervised* outlier model selection. Further, it does not require choosing or tuning any values at test time, and hence is fully automatic. In terms of computation, test-time embedding by $\phi$ (PCA) and regression by $f$ (regression trees) take near-constant time given the small number of meta-features, embedding dimensions, and trees of fixed depth. Moreover, we use meta-features with computational complexity linear in the dataset size as we describe next.

### 3.3 Meta-Features for Outlier Detection

A key part of METAOD is the extraction of meta-features that capture the important characteristics of an arbitrary dataset. Existing outlier detection models have different methodological designs (e.g., density, distance, angle, etc. based) and different assumptions around the topology of outliers (e.g., global, local, clustered). As a result, we expect different models to perform differently depending on the input dataset and the nature of outliers it exhibits—hence no "winner". In our meta-learning approach, the goal is to identify the datasets in the meta-train database that exhibit *similar* characteristics to a given test dataset, and focus on models that do well on those similar datasets. This is akin to recommending to a new user those items liked by similar users.

To this end, we extract meta-features that can be organized into two categories: (1) statistical features, and (2) landmarker features. Broadly speaking, the former captures statistical properties of the underlying data distributions; e.g., min, max, variance, skewness, covariance, etc. of the features and

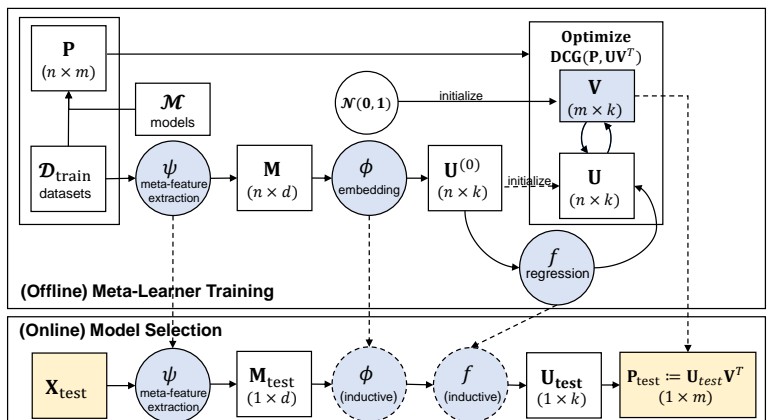

Figure 1: METAOD overview; components that transfer from offline (meta-learning) to online (model selection) phase shown in blue; namely, meta-feature extractors ($\psi$), embedding model ($\phi$), regressor $f$, dataset matrix $\mathbf{U}$, and model matrix $\mathbf{V}$. For the online phase, the input dataset $\mathbf{X}_{\text{test}}$ and the predicted model performance $\mathbf{P}_{\text{test}}$ are denoted in yellow.

feature combinations. (See Appendix B Table 3 for the complete list.) These kinds of meta-features have been commonly used in the AutoML literature [5].

The optimal set of meta-features has been shown to be application-dependent [49]. Therefore, perhaps more important are the landmarker features, which are *problem-specific*, and aim to capture the *outlying* characteristics of a dataset. The idea is to apply a few of the fast, easy-to-construct OD models on a dataset and extract features from ($i$) the structure of the estimated OD model, and ($ii$) its output outlier scores. For the OD-specific landmarkers, we use four OD algorithms: iForest [31], HBOS [17], LODA [34], and PCA [20] (reconstruction error as outlier score). We choose the four OD algorithms due to their efficiency and diversity (as a group). First, they are all fast algorithms and able to handle large, high-dimensional datasets [2]. This makes the meta-feature generation efficient and practical in the real world. Second, these four OD algorithms as a group show decent diversity (i.e., internal detection mechanism) to capture rich outlying characteristics. Consider iForest as an example. It creates a set of what-is-called extremely randomized trees that define the model structure, from which we extract structural features such as average horizontal and vertical tree imbalance. As another example, LODA builds on random-projection histograms from which we extract features such as entropy. In addition, based on the list of outlier scores from these models, we compute features such as dispersion, max consecutive gap in the sorted order, etc. We elaborate on the details of the landmarker features in Appendix B.2.

### 3.4 Meta-Learning Objective and Training

#### 3.4.1 Rank-based Criterion

A typical loss criterion for matrix factorization is the mean squared error (MSE), a.k.a. the Frobenius norm of the error matrix $\mathbf{P} - \mathbf{U}\mathbf{V}^T$. While having nice properties from an optimization perspective, MSE does not (at least directly) concern with the ranking quality. In contrast, our goal is to rank the models for *each* dataset row-wise, as model selection concerns with picking the best possible model to employ. Therefore, we use a rank-based criterion called DCG from the information retrieval literature [21]. For a given ranking, DCG is given as

$$\text{DCG} = \sum_r \frac{b^{rel\_r} - 1}{\log_2(r+1)} \tag{3}$$

where $rel\_r$ depicts the true relevance of the item ranked at the $r$-th position and $b$ is a scalar (typically set to 2). In our setting, we use the performance of a model to reflect its true relevance to a dataset. As such, DCG for dataset $i$ is re-written as

$$\text{DCG}_i = \sum_{j=1}^{m} \frac{b^{\mathbf{P}_{ij}} - 1}{\log_2(1 + \sum_{k=1}^{m} \mathbb{1}[\widehat{\mathbf{P}}_{ij} \leq \widehat{\mathbf{P}}_{ik}])} \tag{4}$$

where $\widehat{\mathbf{P}}_{ij} = \langle \mathbf{U}_i, \mathbf{V}_j \rangle$ is the predicted performance that dictates the ranking order. Intuitively, ranking high-performing models at the top leads to higher DCG, and a larger $b$ increases the emphasis on the quality of models at the higher rank positions.

A challenge with DCG is that it is not differentiable, unlike MSE, as it involves ranking/sorting. Specifically, the sum term in the denominator of Eq. (4) uses the (nonsmooth) indicator function to obtain the position of model $j$ as ranked by the estimated performances. We circumvent this challenge by replacing the indicator function by the (smooth) sigmoid approximation [16] as follows.

$$\text{DCG}_i \approx \text{sDCG}_i = \sum_{j=1}^{m} \frac{b^{\mathbf{P}_{ij}} - 1}{\log_2(1 + \sum_{k=1}^{m} \sigma(\widehat{\mathbf{P}}_{ik} - \widehat{\mathbf{P}}_{ij}))} \tag{5}$$

### 3.4.2 Initialization & Alternating Optimization

Overall we optimize the smoothed criterion, sDCG, over all meta-train datasets $\mathcal{D}_{\text{train}} = \{\mathbf{D}_i\}_{i=1}^{n}$ as

$$\min_{\mathbf{U}, \mathbf{V}} \quad L = -\sum_{i=1}^{n} \text{sDCG}_i(\mathbf{P}_i, \mathbf{U}_i \mathbf{V}^T) , \tag{6}$$

by alternatingly solving for $\mathbf{U}$ as we fix $\mathbf{V}$ (and vice versa) by gradient descent. We initialize $\mathbf{U}$ by leveraging the meta-features, which are embedded to a space with the same size as $\mathbf{U}$. By capturing the latent similarities among the datasets, such an initialization not only accelerates convergence [62] but also facilitates convergence to a better local optimum. $\mathbf{V}$ is initialized from a unit Normal.

As we aim to maximize the total *dataset-wise* DCG, we make a pass over meta-train datasets one by one at each epoch. For brevity, we give the gradients for $\mathbf{U}_i$ and $\mathbf{V}_j$ in Eq.s (7) and (8), respectively.

$$\frac{\partial L}{\partial \mathbf{U}_i} = \ln(2) \sum_{j=1}^{m} \left[ \frac{b^{\mathbf{P}_{ij}} - 1}{\beta_j^i \ln^2(\beta_j^i)} \sum_{k \neq j} \sigma(w_{jk}^i)(1 - \sigma(w_{jk}^i))(\mathbf{V}_k - \mathbf{V}_j) \right] \tag{7}$$

$$\frac{\partial L}{\partial \mathbf{V}_j} = -\ln(2) \sum_{j=1}^{m} \left[ \frac{b^{\mathbf{P}_{ij}} - 1}{\beta_j^i \ln^2(\beta_j^i)} \sum_{k \neq j} \sigma(w_{jk}^i)(1 - \sigma(w_{jk}^i))\mathbf{U}_i \right] \tag{8}$$

where $w_{jk}^i = \langle \mathbf{U}_i, (\mathbf{V}_k - \mathbf{V}_j) \rangle$ and $\beta_j^i = \frac{3}{2} + \sum_{k \neq j} \sigma(w_{jk}^i)$; see derivations in Appendix C.

## 4 Experiments

### 4.1 Experiment Setting

**Model Set and Evaluation.** We pair 8 SOTA OD algorithms and their corresponding hyperparameters to compose a model set $\mathcal{M}$ with 302 unique models. (See Appendix A Table 2 for the complete list.) We evaluate METAOD and the baselines on 2 testbeds introduced below, resp. with 100 and 62 datasets, via cross-validation where datasets are split into meta-train/test in each fold. For each testbed, we first generate the performance matrix $\mathbf{P}$, by evaluating the models from $\mathcal{M}$ against the benchmark datasets in the testbed. For randomized detectors (random-split trees/random projections/etc.), we run five independent trials and record the average performance. For consistency, all models are built using the PyOD library [61] on an Intel i7-9700 @3.00 GHz, 64GB RAM, 8-core workstation. We compare two methods statistically, using the pairwise Wilcoxon signed rank test on performances across datasets (significance level $p < 0.05$).

**Testbed Setup.** Meta-learning works well if a new task can leverage prior knowledge; e.g., mastering motorcycle can benefit from bike riding experience. As such, METAOD relies on the assumption that a newcoming test dataset shares similarity with some meta-train datasets. We create two testbeds with different train/test dataset similarity, to systematically study the effect of task similarity.

1. **Proof-of-Concept (POC) testbed** contains 100 datasets that form clusters of similar datasets, where 5 different detection tasks ("siblings") are created from each one of 20 "mothersets".
2. **Stress Testing (ST) testbed** consists of 62 independent datasets from 3 different public-domain OD dataset repositories , which exhibit relatively lower similarity to one another.

We refer to Appendix E for the complete list of datasets and details on testbed generation. Fig. 2 illustrates the differences between POC and ST testbeds, where the meta-features of their constituting

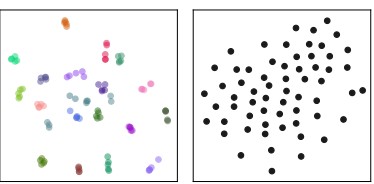

Figure 2: 2-D embedding of datasets in (left) POC and (right) ST. POC exhibits higher task similarity, wherein "siblings" (marked by same color) form clusters. ST contains independent datasets with no apparent clusters.

datasets are embedded to 2-D by t-SNE [48]. By construction, POC consists of clusters and hence exhibits higher task/dataset similarity as compared to ST.

**Baselines.** Being the first work for UOMS, METAOD does not have immediate competing baselines. Therefore we employ simple ideas and tailor some existing methods for comparison. We also create 2 variations of METAOD (marked with †) for ablation analysis.

In Appendix F we give detailed descriptions of all 10 baselines. Briefly, they are organized as follows: ($i$) *no model selection* always employs the same popular model, namely **(1) LOF** [6] or **(2) iForest** [31], or the ensemble of all the models called **(3) Mega Ensemble (ME)**; ($ii$) *simple meta-learners* include **(4)** Global Best **(GB)** that selects the model with the largest avg. performance across meta-train datasets, **(5) ISAC** [23] and **(6)** ARGOSMART **(AS)** [33]; and ($iii$) *optimization-based meta-learners* include **(7)** Supervised Surrogates **(SS)** [55] and **(8) ALORS** [32].

Variants of METAOD are **(9)** †METAOD_C where performance and meta-feature matrices are concatenated as $\mathbf{C} = [\mathbf{P}, \mathbf{M}] \in \mathbb{R}^{n \times (m+d)}$, before factorization, $\mathbf{C} \approx \mathbf{U}\mathbf{V}^T$. Given a test dataset, zero-concatenated meta-features are projected and reconstructed as $[\widehat{\mathbf{P}}_{\text{test}}; \widehat{\mathbf{M}}_{\text{test}}] := [0 \ldots 0; \mathbf{M}_{\text{new}}]\mathbf{V}\mathbf{V}^T$; and **(10)** †METAOD_F where $\mathbf{U}$ is fixed at $\phi(\mathbf{M})$ after the embedding step and only $\mathbf{V}$ is optimized.

Additionally, we report Empirical Upper Bound **(EUB)** (only) for POC, as the performance of the best model on a dataset's 4 "siblings"; this (valuable) information is not available in practice–hence "upper bound". For ST with lower task similarity, we include Random Selection **(RS)** as baseline.

## 4.2 POC Testbed Results

**Testbed Setting**. POC testbed is built to simulate the scenario where there are similar meta-train tasks to a given test task. We use the benchmark datasets[4] by Emmott *et al.* [11], who created "childsets" from 20 independent "mothersets" by sampling. Consequently, the childsets generated from the same motherset using the same generation properties (e.g., the frequency of anomalies) can be deemed as "siblings" with large similarity. We build the POC testbed by using 5 siblings from each motherset, resulting in 100 datasets. We split them into 5 folds for cross-validation, each test fold containing 20 independent childsets without siblings.

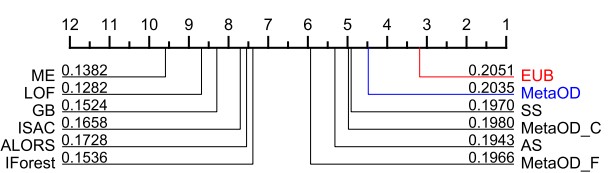

Figure 3: Comparison of avg. rank (lower is better) of methods w.r.t. performance across datasets in POC. Mean AP across datasets (higher is better) shown on lines. METAOD is the top-performing meta-learner, and comparable to EUB.

**Results**. In Fig. 3, we observe that METAOD **is superior to all baseline methods w.r.t. the average rank and mean average precision (MAP), and performs comparably to the Empirical Upper Bound (EUB)**. Table 1 (left) shows that METAOD is the only meta-learner that is not significantly different from both EUB (MAP=0.2051) and the 4-$th$ best model (0.2185). Moreover, METAOD is significantly better than the baselines that do not employ any model selection (LOF (0.1282), iForest (0.1536), and ME (0.1382)), as well as all the other meta-learners including GB (0.1524), ISAC (0.1658) and ALORS (0.1728). For the full POC evaluation, see Appendix G.1.

**Averaging all models (ME) does not lead to good performance** as one may expect. As shown in Fig. 3, ME is the worst baseline by average rank in the POC testbed. Using a single detector, e.g.,

---

[4] https://ir.library.oregonstate.edu/concern/datasets/47429f155

| Ours | Baseline | p-value | Ours | Baseline | p-value |
|---|---|---|---|---|---|
| **MetaOD** | **EUB** | 0.0522 | **MetaOD** | **58-th Best** | 0.0517 |
| **MetaOD** | **4-th Best** | 0.0929 | **MetaOD** | RS | 0.0001 |
| **MetaOD** | LOF | 0.0013 | **MetaOD** | LOF | 0.0001 |
| **MetaOD** | iForest | 0.0090 | **MetaOD** | **iForest** | 0.1129 |
| **MetaOD** | ME | 0.0004 | **MetaOD** | ME | 0.0001 |
| **MetaOD** | GB | 0.0051 | **MetaOD** | GB | 0.0030 |
| **MetaOD** | ISAC | 0.0019 | **MetaOD** | ISAC | 0.0006 |
| **MetaOD** | AS | 0.2959 | **MetaOD** | AS | 0.0009 |
| **MetaOD** | SS | 0.7938 | **MetaOD** | SS | 0.0190 |
| **MetaOD** | ALORS | 0.0025 | **MetaOD** | ALORS | 0.0001 |
| **MetaOD** | MetaOD_C | 0.6874 | **MetaOD** | MetaOD_C | 0.0001 |
| **MetaOD** | MetaOD_F | 0.1165 | **MetaOD** | MetaOD_F | 0.0001 |

Table 1: Pairwise statistical test results between METAOD and baselines by Wilcoxon signed rank test. Statistically better method shown in **bold** (both marked **bold** if no significance). In (left) POC, METAOD is the only meta-learner with no diff. from both EUB and the 4-$th$ best model. In (right) ST, METAOD is the only meta-learner with no statistical diff. from the 58-$th$ best model. It is statistically better than all except iForest.

iForest, is significantly better. This is mainly because some models perform poorly on any given dataset, and ensembling all the models indiscriminately draws overall performance down. Using selective ensembles [36] could be beneficial, however, ensembles of many models are expensive to build in practice. In contrast, METAOD is fast at test time and selects without building any models.

**Meta-learners perform significantly better than methods without model selection.** In particular, four meta-learners (METAOD, SS, METAOD_**C**, METAOD_**F**) significantly outperform single outlier detection methods (LOF and iForest) as well as the Mega Ensemble (ME) that averages all the models. METAOD respectively has 58.74%, 32.48%, and 47.25% higher MAP over LOF, iForest, and ME. These results signify the benefits of model selection.

**Optimization-based meta learners generally perform better than simple meta learners**. Top-3 meta learners by average rank (METAOD, SS, and METAOD_**C**) are all optimization-based and significantly outperform simple meta-learners like ISAC as shown in Fig. 3. Simple meta-learners weigh meta-features equally for task similarity, whereas others learn which meta-features matter (e.g., regression on meta-features), leading to better results. We find that METAOD respectively achieves 33.53%, 22.74%, and 4.73% higher MAP than simple meta-learners including GB, ISAC, and AS.

### 4.3 ST Testbed Results

**Testbed Setting**. When meta-train datasets lack similarity to the test dataset, it is hard to capitalize on prior experience. In the extreme case, meta-learning may not perform better than no-model-selection baselines, e.g., a single detector. To investigate the impact of the train/test similarity on meta-learning performance, we build the ST testbed that consists of 62 public-domain datasets from 3 different repositories (See Appendix E Table 4) with relatively low similarity as shown in Fig. 2. For evaluation on ST, we use leave-one-out cross validation; each time using 61 datasets as meta-train.

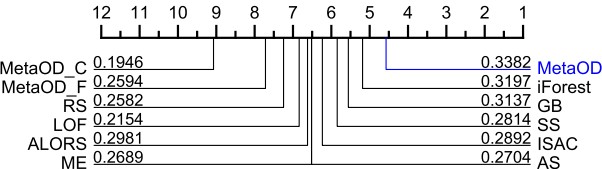

Figure 4: Comparison of avg. rank (lower is better) of methods w.r.t. performance across datasets in ST. Mean AP (higher is better) shown on lines. METAOD outperforms all baselines.

**Results**. For the ST testbed, METAOD **still outperforms all baseline methods w.r.t. average rank and MAP** as shown in Fig. 4. Table 1 (right) shows that METAOD (0.3382) could select, from a pool of 302, the model that is as good as the 58-$th$ best model (top 20%) per dataset (0.3513) in this challenging testbed. The comparable model changes from the 4-$th$ best per dataset in POC to 58-$th$ best in ST, which is expected due to the lower task similarity to leverage in ST. Notably, all other baselines are worse than the 80-$th$ best model with statistical significance. Moreover, METAOD is significantly better than all baselines except iForest. Note that METAOD also significantly outperforms RS, showing that it is able to exploit the meta-train database despite limited task similarity and not simply resorting to random picking. These results suggest that METAOD **is a good choice under various extent of similarity among train/test datasets**. We refer to Appendix G.2 for detailed ST results on individual ST datasets.

**Training stability affects performance for optimization-based methods.** Notably, several optimization-based meta-learners, such as ALORS and METAOD_**C**, do not perform well for ST. We find that the training process of matrix factorization is not stable when latent similarities are

weak. In METAOD, we employ two strategies that help stabilize the training. First, we leverage meta-feature based (rather than random) initialization. Second, we use cyclical learning rates that help escape saddle points for better local optima [43]. Consequently, METAOD (0.3382) significantly outperforms ALORS (0.2981) and METAOD_C (0.1946) with 13.45% and 73.79% higher MAP.

**Global methods outperform local methods under limited task similarity.** In ST, datasets are less similar and simple meta-learners that leverage task similarity locally often perform poorly. For example, AS selects the model based on the 1-NN, and is likely to fail if the most similar meta-train task is still quite dissimilar to the current task. Notably, the global meta-learner GB outperforms AS and ISAC. Note the opposite ordering among these methods in POC as shown in Fig. 3. In short, **effectiveness of simple meta-learners tends to be sensitive to the train/test dataset similarity**, which makes them hard to use in general. In contrast, METAOD performs well in both settings.

### 4.4 Runtime Analysis

Empowered by meta-training, METAOD (meta-feature generation and model selection) takes less than 1 second on most test datasets, as shown in Fig. 5, where it incurs negligible overhead relative to building/training the selected outlier model ($\approx$10% on avg.). Fig. 6 corroborates the statement by showing the comparison on the 10 largest datasets in POC.

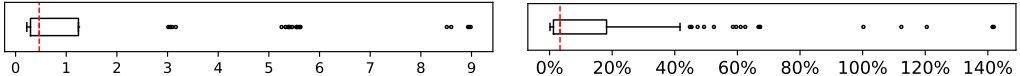

Figure 5: METAOD running time at test time in sec.s (left), and percentage of time relative to building the selected model (right). Notice that it is fast, and incurs negligible computational overhead.

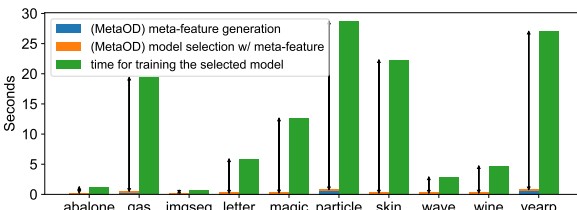

Figure 6: Time for METAOD vs. training of the selected model (on 10 largest datasets in POC). METAOD incurs only negligible overhead (diff. shown w/ black arrows).

Notably, meta-feature extraction may be trivially parallelized whereas the model selection is even faster, e.g., using SUOD [60], effectively taking constant time (See §3.2.2).

## 5 Conclusion

We addressed the unsupervised outlier model selection (UOMS) problem *without relying on any labels, model evaluations or comparisons* for the first time. Our proposed METAOD is a meta-learner, and builds on an extensive pool of historical outlier detection datasets and models. Given a new task, it selects a model based on the past performances of models on similar historical tasks. To effectively capture task similarity, we designed novel problem-specific meta-features. Importantly, METAOD is ($i$) **fully automatic**, requiring no supervision at test time, and ($ii$) **lightweight**, incurring relatively small selection time overhead prior to outlier model building. Extensive experiments on two large testbeds showed that METAOD significantly improves detection performance over always using some of the most popular outlier models as well as several other meta-learners tailored for UOMS.

We open-source[1] METAOD and our meta-learning database for use in practice. We expect meta-learning to become more powerful as the meta-train database grows. Therefore, we also share all our code and testbeds with the community to stimulate further advances in automating UOMS. Future work can address UOMS in the continuous hyperparameter space, leverage self-aware learning [28] and conformal prediction [39] to estimate the confidence in selection, and explore the potential bias and fairness issues in OD model selection [10, 42].

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
