# Supplementary Material: Automatic Unsupervised Outlier Model Selection

*Details on Models, Meta-features, Datasets/Testbeds, Optimization, pseudo code, and Detailed Experiment Result*

## A    METAOD **Model Set**

Model set $\mathcal{M}$ is composed by pairing outlier detection algorithms to distinct hyperparameter choices. Table 2 provides a comprehensive description of models, including 302 unique models composed by 8 popular outlier detection (OD) algorithms. All models and parameters are based on the Python Outlier Detection Toolbox (PyOD)[5].

Table 2: Outlier Detection Models; see hyperparameter definitions from PyOD [61]

| Detection algorithm | Hyperparameter 1 | Hyperparameter 2 | Total |
|---|---|---|---|
| LOF [6] | n_neighbors: $[1, 5, 10, 15, 20, 25, 50, 60, 70, 80, 90, 100]$ | distance: ['manhattan', 'euclidean', 'minkowski'] | 36 |
| kNN [35] | n_neighbors: $[1, 5, 10, 15, 20, 25, 50, 60, 70, 80, 90, 100]$ | method: ['largest', 'mean', 'median'] | 36 |
| OCSVM [37] | nu (train error tol): $[0.1, 0.2, 0.3, 0.4, 0.5, 0.6, 0.7, 0.8, 0.9]$ | kernel: ['linear', 'poly', 'rbf', 'sigmoid'] | 36 |
| COF [45] | n_neighbors: $[3, 5, 10, 15, 20, 25, 50]$ | N/A | 7 |
| ABOD [24] | n_neighbors: $[3, 5, 10, 15, 20, 25, 50, 60, 70, 80, 90, 100]$ | N/A | 7 |
| iForest [31] | n_estimators: $[10, 20, 30, 40, 50, 75, 100, 150, 200]$ | max_features: $[0.1, 0.2, 0.3, 0.4, 0.5, 0.6, 0.7, 0.8, 0.9]$ | 81 |
| HBOS [17] | n_histograms: $[5, 10, 20, 30, 40, 50, 75, 100]$ | tolerance: $[0.1, 0.2, 0.3, 0.4, 0.5]$ | 40 |
| LODA [34] | n_bins: $[10, 20, 30, 40, 50, 75, 100, 150, 200]$ | n_random_cuts: $[5, 10, 15, 20, 25, 30]$ | 54 |
| | | | **302** |

## B    Meta-features

### B.1    Complete List of Meta-features

We summarize the meta-features used by METAOD in Table 3. When applicable, we provide the formula for computing the meta-feature(s) and corresponding variants. Some are based on [49]. Refer to the accompanied code for details.

Specifically, meta-features can be categorized into (1) statistical features, and (2) landmarker features. Broadly speaking, the former captures statistical properties of the underlying data distributions; e.g., min, max, variance, skewness, covariance, etc. of the features and feature combinations. These statistics-based meta-features have been commonly used in the AutoML literature [49].

### B.2    Landmarker Meta-features

In addition to statistical meta-features, we use four OD-specific landmarker algorithms for computing OD-specific landmarker meta-features, iForest [31], HBOS [17], LODA [34], and PCA [20] (reconstruction error as outlier score), to capture outlying characteristics of a dataset. To this end, we first provide a quick overview of each algorithm and then discuss how we are using them for building meta-features. The algorithms are executed with the default parameter. Refer to the attached code for details of meta-feature construction.

**Isolation Forest (iForest)** [31] is a tree-based ensemble method. Specifically, iForest builds a collection of base trees using the subsampled unlabeled data, splitting on (randomly selected) features as nodes. iForest grows internal nodes until the terminal leaves contain only one sample or the predefined max depth is reached. Given the max depth is not set and we have multiple base trees with each leaf containing one sample only, the anomaly score of a sample is the aggregated depth the leaves the sample falls into. The key assumption is that an anomaly is more different than the normal samples, and is, therefore, easier to be "isolated" during the node splitting. Consequently, anomalies are closer to roots with small tree depth. For iForest, we use the balance of base trees (i.e., depth of trees and number of leaves per tree) and additional information (e.g., feature importance of each base tree). It is noted that feature importance information is available for each base tree—we therefore analyze the statistic of mean and max of base tree feature importance. Specifically, the following information of base trees are used:

---
[5]`https://github.com/yzhao062/pyod`

Table 3: Selected meta-features for characterizing an arbitrary dataset. See code for details.

| Name | Formula | Rationale | Variants |
|---|---|---|---|
| Nr instances | n | Speed, Scalability | $\frac{p}{n}, \log(n), \log(\frac{n}{p})$ |
| Nr features | p | Curse of dimensionality | $\log(p)$, % categorical |
| Sample mean | $\mu$ | Concentration | |
| Sample median | $\tilde{X}$ | Concentration | |
| Sample var | $\sigma^2$ | Dispersion | |
| Sample min | $\max_X$ | Data range | |
| Sample max | $\min_X$ | Data range | |
| Sample std | $\sigma$ | Dispersion | |
| Percentile | $P_i$ | Dispersion | q1, q25, q75, q99 |
| Interquartile Range (IQR) | $q75 - q25$ | Dispersion | |
| Normalized mean | $\frac{\mu}{\max_X}$ | Data range | |
| Normalized median | $\frac{\tilde{X}}{\max_X}$ | Data range | |
| Sample range | $\max_X - \min_X$ | Data range | |
| Sample Gini | | Dispersion | |
| Median absolute deviation | $\mathrm{median}(X - \tilde{X})$ | Variability and dispersion | |
| Average absolute deviation | $\mathrm{avg}(X - \tilde{X})$ | Variability and dispersion | |
| Quantile Coefficient Dispersion | $\frac{(q75-q25)}{(q75+q25)}$ | Dispersion | |
| Coefficient of variance | | Dispersion | |
| Outlier outside 1 & 99 | % samples outside 1% or 99% | Basic outlying patterns | |
| Outlier 3 STD | % samples outside $3\sigma$ | Basic outlying patterns | |
| Normal test | If a sample differs from a normal dist. | Feature normality | |
| $k$th moments | | | 5th to 10th moments |
| Skewness | Feature skewness | Feature normality | min, max, $\mu$, $\sigma$, skewness, kurtosis |
| Kurtosis | $\frac{\mu_4}{\sigma^4}$ | Feature normality | min, max, $\mu$, $\sigma$, skewness, kurtosis |
| Correlation | $\rho$ | Feature interdependence | min, max, $\mu$, $\sigma$, skewness, kurtosis |
| Covariance | Cov | Feature interdependence | min, max, $\mu$, $\sigma$, skewness, kurtosis |
| Sparsity | $\frac{\#\text{Unique values}}{n}$ | Degree of discreteness | min, max, $\mu$, $\sigma$, skewness, kurtosis |
| ANOVA p-value | $p_{\mathrm{ANOVA}}$ | Feature redundancy | min, max, $\mu$, $\sigma$, skewness, kurtosis |
| Coeff of variation | $\frac{\sigma_x}{\mu_x}$ | Dispersion | |
| Norm. entropy | $\frac{H(X)}{log_2 n}$ | Feature informativeness | min,max, $\sigma$, $\mu$ |
| Landmarker (HBOS) | See §B.2 | Outlying patterns | Histogram density |
| Landmarker (LODA) | See §B.2 | Outlying patterns | Histogram density |
| Landmarker (PCA) | See §B.2 | Outlying patterns | Explained variance ratio, singular values |
| Landmarker (iForest) | See §B.2 | Outlying patterns | # of leaves, tree depth, feature importance |

- *Tree depth*: min, max, mean, std, skewness, and kurtosis
- *Number of leaves*: min, max, mean, std, skewness, and kurtosis
- *Mean of base tree feature importance*: min, max, mean, std, skewness, and kurtosis
- *Max of base tree feature importance*: min, max, mean, std, skewness, and kurtosis

**Histogram-based Outlier Scores (HBOS)** [17] assumes that each dimension (feature) of the datasets is independent. It builds a histogram on each feature to calculate the density. Given there are $n$ samples and $d$ features, for each histogram from $1...d$, HBOS estimates the sample density using all $n$ samples. Intuitively, the anomaly score of sample $g$ is defined as the sum of log of inverse density. In other words, it can be considered as an aggregation of density estimation on each feature. Obviously, the samples falling in high-density areas are more likely to be normal points and vice versa. The following information is included as part of METAOD:

- *Mean of each histogram (per feature)*: min, max, mean, std, skewness, and kurtosis
- *Max of each histogram (per feature)* : min, max, mean, std, skewness, and kurtosis

**Lightweight on-line detector of anomalies (LODA)** [34] is a fast ensemble-based anomaly detection algorithm. It shares a similar idea as HBOS—"although one one-dimensional histogram is a very weak anomaly detector, their collection yields to a strong detector". Different from HBOS that simply aggregates over all independent histograms, LODA extends the histogram-based model generating $k$ random projection vectors to compress data into one-dimensional space for building histograms. Similar to HBOS, we include in the following information as part of meta-features:

- *Mean of each random projection (per feature)*: min, max, mean, std, skewness, and kurtosis
- *Max of each random projection (per feature)* : min, max, mean, std, skewness, and kurtosis
- *Mean of each histogram (per feature)*: min, max, mean, std, skewness, and kurtosis
- *Max of each histogram (per feature)* : min, max, mean, std, skewness, and kurtosis

**Principal component analysis based outlier detector (PCA)** [20] aims to quantify sample outlyingness by projecting them into lower dimensions through principal component analysis. Since the

number of normal samples is much bigger than the number of outliers, the identified projection matrix is mainly suited for normal samples. Consequently, the reconstruction error of normal samples are smaller than that of outlier samples, which can be used to measure sample outlyingness. For PCA, we include the following information into meta-features:

- *Explained variance ratio on the first three principal components*: The percentage of variance it captures for the top 3 principal components
- *Singular values*: The top 3 singular values generated during SVD process

Additionally, we also leverage the **outlier scores by OD landmarkers** after appropriate scaling, e.g., normalization/standardization.

## C    Gradient Derivations

In this section we provide the details for the gradient derivation of METAOD. It is organized as follow. We first provide a quick overview of gradient derivation in classical recommender systems, and then show the derivation of the rank-based criterion used in METAOD.

### C.1    Background

Given a rating matrix $\mathbf{P} \in \mathbb{R}^{n \times m}$ with $n$ users rating on $m$ items, $\mathbf{P}_{ij}$ denotes $i$th user's rating on the $j$th item in classical recommender system setting. For learning the latent factors in $k$ dimensions, we try to factorize $\mathbf{P}$ into user matrix $\mathbf{U} \in \mathbb{R}^{n \times k}$ and the item matrix $\mathbf{V} \in \mathbb{R}^{d \times k}$ to make $\mathbf{P} \approx \mathbf{U}\mathbf{V}^T$.

In classical matrix factorization setting, some entries of the performance matrix $\mathbf{P}$ is missing. Consequently, one may use stochastic gradient descent to minimize the mean squared error (MSE) between $\mathbf{P}$ and $\mathbf{U}\mathbf{V}^T$ through all non-empty entries. For each rating $\mathbf{P}_{ij}$, the loss $L$ is defined as:

$$L_{i,j} = L(\mathbf{U}_i, \mathbf{V}_j^T) = (\mathbf{P}_{ij} - \mathbf{U}_i\mathbf{V}_j^T)^2 \tag{9}$$

The total loss over all non-empty entries is:

$$L = \sum_{i,j}(\mathbf{P}_{ij} - \mathbf{U}_i\mathbf{V}_j^T)^2 \tag{10}$$

The optimization process iterates over all non-empty entries of the performance matrix $\mathbf{P}$, and updates $\mathbf{U}_i$ and $\mathbf{V}_j$ using the learning rate $\eta$ as:

$$\mathbf{U}_i \leftarrow \mathbf{U}_i - \eta\frac{\partial L}{\partial \mathbf{U}_i} \tag{11}$$

$$\mathbf{V}_j \leftarrow \mathbf{V}_j - \eta\frac{\partial L}{\partial \mathbf{V}_j} \tag{12}$$

### C.2    Gradient Derivation for DCG-based Criterion

As we aim to maximize the total *dataset-wise* DCG, we make a pass over meta-train datasets one by one at each epoch as shown in Algorithm 1. We update $\mathbf{U}_i$ and $\mathbf{V}_j$ by gradient descent as shown below. It is noted that predicted performance of $j$th model on $i$th dataset is defined as the dot product of corresponding dataset and model vector: $\widehat{\mathbf{P}}_{ij} = \mathbf{U}_i\mathbf{V}_j^T$. So Eq. (5) can be rearranged as:

$$-\text{sDCG}_i = \sum_{j=1}^{m} \frac{b^{\mathbf{P}_{ij}} - 1}{\log_2(1 + \sum_{k=1}^{m} \sigma(\widehat{\mathbf{P}}_{ik} - \widehat{\mathbf{P}}_{ij}))}$$

$$= \ln(2) \sum_{j=1}^{m} \frac{b^{\mathbf{P}_{ij}} - 1}{\ln(1 + \sum_{k=1}^{m} \sigma(\mathbf{U}_i \mathbf{V}_k^T - \mathbf{U}_i \mathbf{V}_j^T))}$$

$$= \ln(2) \sum_{j=1}^{m} \frac{b^{\mathbf{P}_{ij}} - 1}{\ln(1 + \sigma(0) + \sum_{k \neq j} \sigma(\mathbf{U}_i \mathbf{V}_k^T - \mathbf{U}_i \mathbf{V}_j^T))}$$

$$= \ln(2) \sum_{j=1}^{m} \frac{b^{\mathbf{P}_{ij}} - 1}{\ln(\frac{3}{2} + \sum_{k \neq j} \sigma(\mathbf{U}_i \mathbf{V}_k^T - \mathbf{U}_i \mathbf{V}_j^T))}$$

$$= \ln(2) \sum_{j=1}^{m} \frac{b^{\mathbf{P}_{ij}} - 1}{\ln(\frac{3}{2} + \sum_{k \neq j} \sigma(\mathbf{U}_i \mathbf{V}_k^T - \mathbf{U}_i \mathbf{V}_j^T))} \tag{13}$$

$$\tag{14}$$

We compute the gradient of $\mathbf{U}_i$ and $\mathbf{V}_j^T$ as the partial derivative of $-\text{sDCG}_i$ as shown in Eq. (13). To ease the notation, we define

$$w_{jk}^i = \mathbf{U}_i \mathbf{V}_k^T - \mathbf{U}_i \mathbf{V}_j^T = \langle \mathbf{U}_i, (\mathbf{V}_k - \mathbf{V}_j) \rangle \tag{15}$$

$$\beta_j^i = \frac{3}{2} + \sum_{k \neq j} \sigma(\mathbf{U}_i \mathbf{V}_k^T - \mathbf{U}_i \mathbf{V}_j^T) = \frac{3}{2} + \sum_{k \neq j} \sigma(w_{jk}^i) \tag{16}$$

By plugging Eq. (15) and (16) back into Eq. (13), it is simplified into

$$-\text{sDCG}_i = \ln(2) \sum_{j=1}^{m} \frac{b^{\mathbf{P}_{ij}} - 1}{\ln(\frac{3}{2} + \sum_{k \neq j} \sigma(\mathbf{U}_i \mathbf{V}_k^T - \mathbf{U}_i \mathbf{V}_j^T))} = \ln(2) \sum_{j=1}^{m} \frac{b^{\mathbf{P}_{ij}} - 1}{\ln(\beta_j^i)} \tag{17}$$

$$\tag{18}$$

We then obtain the gradients of $\mathbf{U}_i$ and $\mathbf{V}_j^T$ as follows:

$$\frac{\partial L}{\partial \mathbf{U}_i} = \frac{\partial(-\text{sDCG}_i)}{\partial \mathbf{U}_i} = \ln(2) \frac{\partial \left( \sum_{j=1}^{m} \frac{b^{\mathbf{P}_{ij}} - 1}{\ln(\beta_j^i)} \right)}{\partial \mathbf{U}_i}$$

$$\tag{19}$$

$$= \ln(2) \sum_{j=1}^{m} \left[ \frac{b^{\mathbf{P}_{ij}} - 1}{\beta_j^i \ln^2(\beta_j^i)} \sum_{k \neq j} \sigma(w_{jk}^i)(1 - \sigma(w_{jk}^i))(\mathbf{V}_k - \mathbf{V}_j) \right] \tag{20}$$

$$\frac{\partial L}{\partial \mathbf{V}_j} = \frac{\partial(-\text{sDCG}_i)}{\partial \mathbf{V}_j} = \ln(2) \frac{\partial \left( \sum_{j=1}^{m} \frac{b^{\mathbf{P}_{ij}} - 1}{\ln(\beta_j^i)} \right)}{\partial \mathbf{U}_i}$$

$$\tag{21}$$

$$= -\ln(2) \sum_{j=1}^{m} \left[ \frac{b^{\mathbf{P}_{ij}} - 1}{\beta_j^i \ln^2(\beta_j^i)} \sum_{k \neq j} \sigma(w_{jk}^i)(1 - \sigma(w_{jk}^i))\mathbf{U}_i \right] \tag{22}$$

# D  METAOD **Pseudo Code**

Algorithm 1 provides detailed steps of METAOD, for both offline (meta-learning) and online (model selection) stages.

---

**Algorithm 1** METAOD: Offline and Online Phases

---

**Input:** (Offline) meta-train database $\mathcal{D}_{\text{train}}$, model set $\mathcal{M}$, latent dimension $k$; (Online) new OD dataset $\mathbf{X}_{\text{test}}$

**Output:** (Offline) Meta-learner for OD model selection; (Online) Selected model for $\mathbf{X}_{\text{test}}$

---

  ▶ (Offline) OD Meta-learner Training **(§3.2.1)**
1: Train & evaluate $\mathcal{M}$ on $\mathcal{D}_{\text{train}}$ to get performance matrix $\mathbf{P}$
2: Extract meta-features **(§3.3)**, $\mathbf{M} := \psi(\{\mathbf{X}_1, \ldots, \mathbf{X}_n\})$
3: Init. $\mathbf{U}^{(0)}$ by embedding meta-features, $\mathbf{U}^{(0)} := \phi(\mathbf{M}; k)$
4: Init. $\mathbf{V}^{(0)}$ by standard normal dist.n $\mathbf{V}^{(0)} \sim \mathcal{N}(0, 1)$
5: **while** not converged **do**           ▶ alternate. opt. by SGD, **(§3.4)**
6:   Shuffle dataset order in $\mathcal{D}_{\text{train}}$
7:   **for** $i = 1, \ldots, n$ **do**
8:    Update $\mathbf{U}_i$ by Eq. (7)
9:    **for** $j = 1, \ldots, m$ **do**
10:     Update $\mathbf{V}_j$ by Eq. (8)
11:    **end for**
12:   **end for**
13: **end while**
14: Train $f$ regressing $\phi(\mathbf{M}; k)$ onto $\mathbf{U}$ (at convergence)
15: **Save** extractors $\psi$, embed. $\phi$, regressor $f$, $\mathbf{V}$ (at conv.)

---

  ▶ (Online) OD Model Selection **(§3.2.2)**
16: Extract meta-features, $\mathbf{M}_{\text{test}} := \psi(\mathbf{X}_{\text{test}})$
17: Get latent vector after embedding, $\mathbf{U}_{\text{test}} := f(\phi(\mathbf{M}_{\text{test}}))$
18: Predict model set performance, $\mathbf{P}_{\text{test}} := \mathbf{U}_{\text{test}}\mathbf{V}^T$
19: **Return** $\arg\max_j \mathbf{P}_{\text{test}}(j)$ as the selected model for $\mathbf{X}_{\text{test}}$

---

# E  Dataset Description and Testbed Setup

## E.1  POC Testbed Setup

POC testbed is built to simulate the testbed when meta-train and test datasets come from similar distribution. Model selection on test data can therefore benefit from the prior experience on the train set. For this purpose, we use the benchmark datasets[6] [11]. In short, they adapt 19 datasets from UCI repository and also create a synthetic dataset to make a pool of 20 "mothersets". For each motherset, they first separate anomalies from normal points, and then generate "childsets" from the motherset by sampling and controlling outlying properties: (i) point difficulty; (ii) relative frequency, i.e., the number of anomalies; (iii) clusteredness and (iv) feature irrelevance. Taking this approach, the childsets generated from the same motherset with the same properties are deemed to be "siblings" with high similarity. Refer to the original paper for details of the data generation process.

We build the POC testbed by selecting five siblings from each motherset, resulting in 100 datasets. For robustness, we split the 100 datasets into 5 folds for cross-validation. Each fold contains 20 independent datasets with no siblings, and the corresponding train set (80 datasets) contain their siblings. Refer to the code for the 100 randomly selected childsets for POC testbed.

## E.2  ST Testbed Setup

Different from the setting of POC, ST testbed aims to test out METAOD's performance *in the wild*, i.e. when train and test datasets are all independent with limited similarity.

---

[6]`https://ir.library.oregonstate.edu/concern/datasets/47429f155`

To build the ST testbed, we combine the datasets from three different sources resulting in 62 independent datasets as shown in Table 4: (i) 23 datasets from ODDS Library[7]; (ii) 19 datasets DAMI datasets [8][8] as well as (iii) 20 benchmark datasets[6] [11] used in POC. For ST testbed, we run leave-one-out cross validation. That is, each time 61 datasets are used for meta-train and the remaining one for test.

# F  Baselines: Detailed Description

The 10 baseline methods are organized into three categories:

*(i) No model selection* always employs either the same model, specifically the popular LOF or iForest detector, or the ensemble of all the models:

- **Local outlier factor (LOF)** [6] is a popular OD method that measures a sample's deviation in the local region regarding its neighbors.
- **Isolation Forest (iForest)** [31] is a SOTA tree ensemble that measures the difficulty of "isolating" a sample via randomized splits in feature space.
- **Mega Ensemble (ME)** averages outlier scores from the 302 models for a given dataset. ME does not perform model selection but rather uses *all* the models.

*(ii) Simple meta-learners* pick the generally well-performing model, globally or locally:

- **Global Best (GB)** is the *simplest meta-learner* that selects the model with the largest avg. performance across all train datasets, *without* using meta-features.
- **ISAC** [23] clusters the meta-train datasets based on meta-features. Given a new dataset, it identifies its closest cluster and selects the best model with largest avg. performance on the cluster's datasets.
- **ARGOSMART (AS)** [33] finds the closest meta-train dataset (1NN) to a given test dataset, based on meta-feature similarity, and selects the model with the best performance on the 1NN dataset.

*(iii) Optimization-based meta-learners* *learn* meta-feature based task similarities toward optimizing performance estimates:

- **Supervised Surrogates (SS)** [55]: Given the meta-train datasets, it directly maps the meta-features onto model performances by regression.
- **ALORS** [32] factorizes the performance matrix to latent factors, and estimates performance as dot product of the latent factors. A non-linear regressor maps meta-features onto latent factors.
- †METAOD_C is a variant: performance and meta-feature matrices are concatenated as $\mathbf{C} = [\mathbf{P}, \mathbf{M}] \in \mathbb{R}^{n \times (m+d)}$, before factorization, $\mathbf{C} \approx \mathbf{U}\mathbf{V}^T$. Given a new dataset, zero-concatenated meta-features are projected and reconstructed as $[\widehat{\mathbf{P}}_{\text{new}}; \widehat{\mathbf{M}}_{\text{new}}] = [0 \ldots 0; \mathbf{M}_{\text{new}}]\mathbf{V}\mathbf{V}^T$.
- †METAOD_F is a variant, where $\mathbf{U}$ is fixed at $\phi(\mathbf{M})$ after the embedding step; only $\mathbf{V}$ is optimized.

Additionally, we report **Empirical Upper Bound (EUB)** (only applicable to POC): Recall that each POC dataset has 4 "siblings" from the same motherset with similar outlying properties. We consider the performance of the best model on a dataset's "siblings" as its EUB, as siblings provide significant information as to which models are suitable. Note that this (valuable) information is generally not available in practice—hence serves as an upper bound. As for ST with lower task similarity, we include **Random Selection (RS)** as a baseline to quantify how the methods compare to random.

# G  Additional Experiment Results

## G.1  Experiment Results for POC Testbed

We present the performances of compared methods in Table 5, and hypothesis test results in Table 6. It is noted these results are averaged across five folds. The results shows that METAOD achieves the best MAP among all meta-learners.

---

[7] http://odds.cs.stonybrook.edu
[8] http://www.dbs.ifi.lmu.de/research/outlier-evaluation/DAMI

Table 4: ST testbed composed by ODDS library (23 datasets), DAMI library (19 datasets), and Emmott benchmark (20 datasets)

| | Data | Pts | Dim | % Outlier |
|---|---|---|---|---|
| 1 | annthyroid (ODDS) | 7200 | 6 | 7.4167 |
| 2 | arrhythmia (ODDS) | 452 | 274 | 14.6018 |
| 3 | breastw (ODDS) | 683 | 9 | 34.9927 |
| 4 | glass (ODDS) | 214 | 9 | 4.2056 |
| 5 | ionosphere (ODDS) | 351 | 33 | 35.8974 |
| 6 | letter (ODDS) | 1600 | 32 | 6.25 |
| 7 | lympho (ODDS) | 148 | 18 | 4.0541 |
| 8 | mammography (ODDS) | 11183 | 6 | 2.325 |
| 9 | mnist (ODDS) | 7603 | 100 | 9.2069 |
| 10 | musk (ODDS) | 3062 | 166 | 3.1679 |
| 11 | optdigits (ODDS) | 5216 | 64 | 2.8758 |
| 12 | pendigits (ODDS) | 6870 | 16 | 2.2707 |
| 13 | pima (ODDS) | 768 | 8 | 34.8958 |
| 14 | satellite (ODDS) | 6435 | 36 | 31.6395 |
| 15 | satimage-2 (ODDS) | 5803 | 36 | 1.2235 |
| 16 | shuttle (ODDS) | 49097 | 9 | 7.1511 |
| 17 | smtp_n (ODDS) | 95156 | 3 | 0.0315 |
| 18 | speech (ODDS) | 3686 | 400 | 1.6549 |
| 19 | thyroid (ODDS) | 3772 | 6 | 2.4655 |
| 20 | vertebral (ODDS) | 240 | 6 | 12.5 |
| 21 | vowels (ODDS) | 1456 | 12 | 3.4341 |
| 22 | wbc (ODDS) | 378 | 30 | 5.5556 |
| 23 | wine (ODDS) | 129 | 13 | 7.7519 |
| 24 | Annthyroid (DAMI) | 7129 | 21 | 7.4905 |
| 25 | Arrhythmia (DAMI) | 450 | 259 | 45.7778 |
| 26 | Cardiotocography (DAMI) | 2114 | 21 | 22.0435 |
| 27 | HeartDisease (DAMI) | 270 | 13 | 44.4444 |
| 28 | Hepatitis (DAMI) | 80 | 19 | 16.25 |
| 29 | InternetAds (DAMI) | 1966 | 1555 | 18.7182 |
| 30 | PageBlocks (DAMI) | 5393 | 10 | 9.4567 |
| 31 | Pima (DAMI) | 768 | 8 | 34.8958 |
| 32 | SpamBase (DAMI) | 4207 | 57 | 39.9097 |
| 33 | Stamps (DAMI) | 340 | 9 | 9.1176 |
| 34 | Wilt (DAMI) | 4819 | 5 | 5.3331 |
| 35 | ALOI (DAMI) | 49534 | 27 | 3.0444 |
| 36 | Glass (DAMI) | 214 | 7 | 4.2056 |
| 37 | PenDigits (DAMI) | 9868 | 16 | 0.2027 |
| 38 | Shuttle (DAMI) | 1013 | 9 | 1.2833 |
| 39 | Waveform (DAMI) | 3443 | 21 | 2.9044 |
| 40 | WBC (DAMI) | 223 | 9 | 4.4843 |
| 41 | WDBC (DAMI) | 367 | 30 | 2.7248 |
| 42 | WPBC (DAMI) | 198 | 33 | 23.7374 |
| 43 | abalone_1231 (Emmott) | 1986 | 15 | 5.0352 |
| 44 | comm.and.crime_0936 (Emmott) | 910 | 404 | 1.0989 |
| 45 | concrete_1096 (Emmott) | 468 | 32 | 1.0684 |
| 46 | fault_0246 (Emmott) | 278 | 38 | 17.9856 |
| 47 | gas_0321 (Emmott) | 6000 | 128 | 0.1 |
| 48 | imgseg_1526 (Emmott) | 1320 | 25 | 10 |
| 49 | landsat_1761 (Emmott) | 230 | 36 | 10 |
| 50 | letter.rec_1666 (Emmott) | 4089 | 23 | 10.0024 |
| 51 | magic.gamma_1411 (Emmott) | 6000 | 22 | 5 |
| 52 | opt.digits_1316 (Emmott) | 3180 | 248 | 5 |
| 53 | pageb_0126 (Emmott) | 733 | 14 | 16.2347 |
| 54 | particle_1336 (Emmott) | 6000 | 200 | 5 |
| 55 | shuttle_0071 (Emmott) | 6000 | 20 | 16.3167 |
| 56 | skin_1706 (Emmott) | 6000 | 4 | 10 |
| 57 | spambase_0681 (Emmott) | 2522 | 57 | 0.5155 |
| 58 | synthetic_1786 (Emmott) | 329 | 14 | 10.0304 |
| 59 | wave_0661 (Emmott) | 3024 | 21 | 0.5291 |
| 60 | wine_0611 (Emmott) | 3720 | 24 | 0.5108 |
| 61 | yeast_1221 (Emmott) | 926 | 8 | 5.0756 |
| 62 | yearp_0231 (Emmott) | 6000 | 202 | 48.6 |

Table 5: Method evaluation in POC testbed (average precision). The most performing method is highlighted in **bold**. The rank is provided in parenthesis (lower ranks denote better performance). METAOD achieves the best average precision and average rank among all meta-learners.

| Dataset | LOF | iForest | ME | GB | ISAC | AS | SS | ALORS | MetaOD_C | MetaOD_F | MetaOD | EUB |
|---|---|---|---|---|---|---|---|---|---|---|---|---|
| abalone | 0.0812 (12) | 0.1679 (10) | 0.1441 (11) | 0.1738 (9) | 0.192 (6) | 0.229 (3) | 0.224 (4) | 0.1747 (8) | 0.1815 (7) | 0.2141 (5) | 0.2329 (2) | **0.2394 (1)** |
| comm.and.crime | 0.0839 (10) | 0.0913 (8) | 0.0797 (11) | 0.0855 (9) | 0.0638 (12) | 0.1001 (5) | 0.1122 (2) | 0.099 (7) | 0.1079 (3) | 0.0999 (6) | 0.1072 (4) | **0.1156 (1)** |
| concrete | 0.0297 (2) | 0.0279 (3) | **0.0298 (1)** | 0.0251 (5) | 0.0224 (7) | 0.022 (8) | 0.0279 (3) | 0.0236 (6) | 0.0131 (12) | 0.0198 (9) | 0.0185 (10) | 0.0147 (11) |
| fault | 0.1898 (12) | 0.3755 (7) | 0.323 (10) | 0.3735 (8) | 0.197 (11) | 0.3949 (2) | 0.3778 (5) | 0.3723 (9) | **0.4217 (1)** | 0.3813 (4) | 0.3768 (6) | 0.3899 (3) |
| gas | 0.0193 (5) | 0.0031 (11) | 0.0083 (8) | 0.0033 (10) | 0.0059 (9) | **0.0481 (1)** | 0.0152 (6) | 0.0024 (12) | 0.0392 (2) | 0.013 (7) | 0.0229 (4) | 0.0387 (3) |
| imgseg | 0.1153 (12) | 0.3618 (6) | 0.2586 (11) | 0.3598 (9) | 0.3659 (5) | 0.3514 (10) | 0.3891 (4) | 0.3605 (8) | 0.3612 (7) | 0.3989 (3) | 0.408 (2) | **0.4166 (1)** |
| landsat | 0.1644 (4) | 0.1306 (9) | 0.131 (8) | 0.13 (10) | 0.1071 (12) | 0.1578 (5) | 0.138 (7) | 0.1111 (11) | **0.1844 (1)** | 0.1562 (6) | 0.1784 (3) | **0.1844 (1)** |
| letter.rec | 0.1529 (7) | 0.0986 (11) | 0.112 (9) | 0.0991 (10) | 0.0946 (12) | 0.222 (2) | 0.218 (5) | 0.1168 (8) | 0.2216 (3) | **0.2229 (1)** | 0.2179 (6) | 0.2216 (3) |
| magic.gamma | 0.1152 (9) | 0.1303 (4) | 0.1104 (10) | 0.1314 (3) | 0.1096 (11) | 0.1319 (2) | 0.1299 (5) | 0.1294 (6) | 0.102 (12) | 0.1255 (8) | 0.1263 (7) | **0.1351 (1)** |
| opt.digits | 0.0662 (9) | 0.0668 (7) | 0.0603 (12) | 0.0665 (8) | 0.0742 (3) | 0.0795 (2) | 0.0689 (6) | 0.0606 (11) | 0.0705 (4) | 0.066 (10) | 0.0701 (5) | **0.0803 (1)** |
| pageb | 0.3956 (11) | 0.4581 (6) | 0.3801 (12) | 0.4574 (7) | 0.4384 (9) | 0.4829 (3) | 0.4616 (5) | 0.4621 (4) | 0.4281 (10) | 0.4498 (8) | 0.4898 (2) | **0.4939 (1)** |
| particle | 0.0546 (12) | 0.0782 (5) | 0.0626 (11) | 0.0746 (8) | 0.0761 (6) | **0.1 (1)** | 0.0739 (9) | 0.0683 (10) | 0.0867 (4) | 0.0757 (7) | 0.0892 (3) | 0.0982 (2) |
| shuttle | 0.2015 (11) | 0.2058 (9) | 0.1935 (12) | 0.2056 (10) | 0.2961 (5) | 0.3165 (3) | 0.2711 (7) | 0.2105 (8) | 0.3165 (3) | 0.2932 (6) | **0.3225 (1)** | 0.3185 (2) |
| skin | 0.1161 (9) | 0.0926 (11) | 0.0995 (10) | 0.0911 (12) | 0.1814 (6) | 0.165 (8) | 0.2408 (4) | 0.1737 (7) | **0.2808 (1)** | 0.2808 (1) | 0.2808 (1) | 0.2278 (5) |
| spambase | 0.0187 (12) | 0.0713 (9) | 0.0571 (11) | 0.0706 (10) | 0.0873 (6) | 0.0744 (8) | **0.1292 (1)** | 0.0757 (7) | 0.0981 (4) | 0.112 (2) | 0.0942 (5) | 0.0982 (3) |
| synthetic | 0.1233 (4) | 0.1226 (5) | 0.1157 (8) | 0.1132 (11) | **0.154 (1)** | 0.1218 (6) | 0.1151 (9) | 0.1147 (10) | 0.1468 (3) | 0.1182 (7) | 0.1483 (2) | 0.1046 (12) |
| wave | 0.0577 (9) | 0.0114 (12) | 0.0297 (10) | 0.0117 (11) | 0.2925 (8) | 0.3413 (5) | **0.3486 (1)** | 0.3244 (7) | **0.3486 (1)** | 0.344 (4) | 0.3365 (6) | 0.3185 (2) |
| wine | 0.0082 (11) | 0.0087 (5) | 0.0084 (10) | 0.0085 (8) | 0.0085 (8) | 0.0073 (12) | 0.0087 (5) | 0.0124 (2) | 0.0097 (3) | 0.0086 (7) | 0.0088 (4) | **0.0129 (1)** |
| yeast | 0.0813 (2) | 0.0781 (4) | 0.073 (8) | 0.0762 (6) | 0.0796 (3) | 0.068 (11) | 0.0781 (4) | 0.0693 (9) | **0.0885 (1)** | 0.067 (12) | 0.0733 (7) | 0.0688 (10) |
| yearp | 0.4894 (5) | 0.4911 (4) | 0.4862 (7) | 0.4913 (3) | 0.4703 (12) | 0.4716 (10) | 0.4916 (2) | 0.4891 (6) | 0.4716 (10) | 0.4741 (9) | 0.4801 (8) | **0.4937 (1)** |
| **average** | 0.1282 (8.7) | 0.1536 (7.4) | 0.1382 (9.7) | 0.1524 (8.35) | 0.1658 (7.6) | 0.1943 (5.49) | 0.197 (4.95) | 0.1728 (7.6) | 0.198 (4.9) | 0.1966 (5.95) | 0.2035 (4.48) | **0.2051 (3.2)** |
| **STD** | 0.1219 | 0.1487 | 0.1294 | 0.1489 | 0.1386 | 0.1491 | 0.1486 | 0.1487 | 0.1485 | 0.1530 | 0.1587 | 0.156 |

Table 6: Pairwise statistical test results between METAOD and baselines by Wilcoxon signed rank test in POC. Statistically better method shown in **bold** (both marked **bold** if no significance). METAOD related pairs are surrounded by rectangles. METAOD (MAP=0.2035) is statistically significantly better than baselines including LOF (0.1282), iForest (0.1536), ME (0.1382), GB (0.1524), ISAC (0.1658), and ALORS (0.1728), and comparable to EUB (0.2051), the empirical upper bound.

| Method 1 | Method 2 | p-value |
|---|---|---|
| **LOF (0.1282)** | **IForest (0.1536)** | 0.3135 |
| **LOF (0.1282)** | **ME (0.1382)** | 0.433 |
| **LOF (0.1282)** | **GB (0.1524)** | 0.4553 |
| **LOF (0.1282)** | **ISAC (0.1658)** | 0.1005 |
| LOF (0.1282) | **AS (0.1943)** | 0.0025 |
| LOF (0.1282) | **SS (0.197)** | 0.0045 |
| **LOF (0.1282)** | **ALORS (0.1728)** | 0.062 |
| LOF (0.1282) | **MetaOD_C (0.198)** | 0.001 |
| LOF (0.1282) | **MetaOD_F (0.1966)** | 0.01 |
| LOF (0.1282) | **MetaOD (0.2035)** | 0.0013 |
| LOF (0.1282) | **EUB (0.2051)** | 0.0007 |
| **IForest (0.1536)** | ME (0.1382) | 0.029 |
| **IForest (0.1536)** | GB (0.1524) | 0.0365 |
| **IForest (0.1536)** | **ISAC (0.1658)** | 0.7369 |
| IForest (0.1536) | **AS (0.1943)** | 0.008 |
| IForest (0.1536) | **SS (0.197)** | 0.0012 |
| **IForest (0.1536)** | **ALORS (0.1728)** | 0.6274 |
| IForest (0.1536) | **MetaOD_C (0.198)** | 0.0276 |
| IForest (0.1536) | **MetaOD_F (0.1966)** | 0.0276 |
| IForest (0.1536) | **MetaOD (0.2035)** | 0.009 |
| IForest (0.1536) | **EUB (0.2051)** | 0.001 |
| **ME (0.1382)** | **GB (0.1524)** | 0.0569 |
| **ME (0.1382)** | **ISAC (0.1658)** | 0.156 |
| ME (0.1382) | **AS (0.1943)** | 0.0005 |
| ME (0.1382) | **SS (0.197)** | 0.0002 |
| ME (0.1382) | **ALORS (0.1728)** | 0.009 |
| ME (0.1382) | **MetaOD_C (0.198)** | 0.0008 |
| ME (0.1382) | **MetaOD_F (0.1966)** | 0.0012 |
| ME (0.1382) | **MetaOD (0.2035)** | 0.0004 |
| ME (0.1382) | **EUB (0.2051)** | 0.0004 |
| **GB (0.1524)** | **ISAC (0.1658)** | 0.7172 |
| GB (0.1524) | **AS (0.1943)** | 0.0032 |
| GB (0.1524) | **SS (0.197)** | 0.0002 |
| **GB (0.1524)** | **ALORS (0.1728)** | 0.3703 |
| GB (0.1524) | **MetaOD_C (0.198)** | 0.0169 |
| GB (0.1524) | **MetaOD_F (0.1966)** | 0.0111 |
| GB (0.1524) | **MetaOD (0.2035)** | 0.0051 |
| GB (0.1524) | **EUB (0.2051)** | 0.0008 |
| ISAC (0.1658) | **AS (0.1943)** | 0.0169 |
| ISAC (0.1658) | **SS (0.197)** | 0.0051 |
| **ISAC (0.1658)** | **ALORS (0.1728)** | 0.6542 |
| **ISAC (0.1658)** | **MetaOD_C (0.198)** | 0.062 |
| ISAC (0.1658) | **MetaOD_F (0.1966)** | 0.008 |
| ISAC (0.1658) | **MetaOD (0.2035)** | 0.0019 |
| ISAC (0.1658) | **EUB (0.2051)** | 0.0015 |
| **AS (0.1943)** | **SS (0.197)** | 0.9702 |
| **AS (0.1943)** | ALORS (0.1728) | 0.0206 |
| **AS (0.1943)** | **MetaOD_C (0.198)** | 0.8446 |
| **AS (0.1943)** | **MetaOD_F (0.1966)** | 0.3135 |
| **AS (0.1943)** | **MetaOD (0.2035)** | 0.2959 |
| AS (0.1943) | **EUB (0.2051)** | 0.0304 |
| **SS (0.197)** | ALORS (0.1728) | 0.0003 |
| **SS (0.197)** | **MetaOD_C (0.198)** | 0.8092 |
| **SS (0.197)** | **MetaOD_F (0.1966)** | 0.4553 |
| **SS (0.197)** | **MetaOD (0.2035)** | 0.7938 |
| **SS (0.197)** | **EUB (0.2051)** | 0.099 |
| ALORS (0.1728) | **MetaOD_C (0.198)** | 0.0276 |
| ALORS (0.1728) | **MetaOD_F (0.1966)** | 0.0137 |
| ALORS (0.1728) | **MetaOD (0.2035)** | 0.0025 |
| ALORS (0.1728) | **EUB (0.2051)** | 0.0006 |
| **MetaOD_C (0.198)** | **MetaOD_F (0.1966)** | 0.7475 |
| **MetaOD_C (0.198)** | **MetaOD (0.2035)** | 0.6874 |
| **MetaOD_C (0.198)** | **EUB (0.2051)** | 0.1773 |
| **MetaOD_F (0.1966)** | **MetaOD (0.2035)** | 0.1165 |
| MetaOD_F (0.1966) | **EUB (0.2051)** | 0.0251 |
| **MetaOD (0.2035)** | **EUB (0.2051)** | 0.0522 |

## G.2 Experiment Results for ST Testbed

We present the method performance in Table 7, and hypothesis test result in Table 8. Among all meta-learners, METAOD shows the best MAP.

Table 7: Method evaluation in ST testbed (average precision). The most performing method is highlighted in **bold**. The rank is provided in parenthesis (lower ranks denote better performance). METAOD achieves the best average precision and average rank among all meta-learners.

| Datasets | LOF | iForest | ME | GB | ISAC | AS | SS | ALORS | MetaOD_c | MetaOD_F | RS | MetaOD |
|---|---|---|---|---|---|---|---|---|---|---|---|---|
| abalone | 0.092 (10) | 0.1654 (2) | 0.1338 (6) | 0.1584 (3) | 0.15 (4) | 0.1232 (9) | **0.1688 (1)** | 0.1316 (7) | 0.0737 (12) | 0.1274 (8) | 0.092 (10) | 0.1355 (5) |
| ALOI | 0.1424 (2) | 0.0333 (6) | 0.0284 (11) | 0.0333 (6) | 0.0329 (9) | **0.5714 (1)** | 0.042 (4) | 0.0282 (12) | 0.0335 (5) | 0.0297 (10) | 0.0491 (3) | 0.0333 (6) |
| annthyroid | 0.1522 (11) | 0.2828 (7) | 0.3177 (6) | 0.3399 (5) | 0.397 (2) | **0.8089 (1)** | 0.3624 (4) | 0.2716 (8) | 0.0605 (12) | 0.196 (10) | 0.2384 (9) | 0.3724 (3) |
| Annthyroid2 | 0.1351 (2) | 0.1198 (5) | 0.1173 (6) | 0.0998 (8) | **0.1353 (1)** | 0.109 (7) | 0.0998 (8) | 0.127 (4) | 0.0837 (11) | 0.0715 (12) | 0.1283 (3) | 0.0998 (8) |
| Arrhythmia | 0.7435 (5) | 0.7615 (2) | 0.6643 (9) | **0.7622 (1)** | 0.7471 (4) | 0.0478 (12) | 0.7134 (8) | 0.7252 (7) | 0.3496 (11) | 0.7273 (6) | 0.3909 (10) | 0.7606 (3) |
| arrhythmia2 | 0.3925 (9) | 0.463 (4) | 0.1833 (10) | 0.4664 (2) | 0.4239 (7) | 0.0396 (12) | 0.3949 (8) | 0.4323 (5) | 0.1833 (10) | 0.4269 (6) | **0.6772 (1)** | 0.4664 (2) |
| breastw | 0.2822 (12) | 0.9695 (3) | 0.9716 (2) | 0.9684 (4) | 0.9564 (7) | 0.3742 (11) | 0.9504 (8) | 0.9615 (6) | 0.817 (9) | **0.9741 (1)** | 0.8022 (10) | 0.9632 (5) |
| Cardio | 0.2802 (10) | 0.4454 (4) | 0.4418 (5) | 0.4187 (7) | 0.3299 (9) | 0.2313 (12) | **0.5306 (1)** | 0.2325 (11) | 0.4989 (2) | 0.4785 (3) | 0.3611 (8) | 0.4233 (6) |
| comm | 0.1104 (2) | 0.0418 (6) | 0.0486 (5) | 0.0251 (9) | 0.0397 (7) | **0.6461 (1)** | 0.073 (3) | 0.0289 (8) | 0.01 (12) | 0.0153 (10) | 0.0153 (10) | 0.0509 (4) |
| concrete | 0.0951 (3) | 0.0502 (5) | 0.0153 (12) | 0.0455 (9) | 0.0412 (10) | **0.1347 (1)** | 0.0493 (6) | 0.0217 (11) | 0.0471 (8) | 0.0493 (6) | **0.1347 (1)** | 0.0508 (4) |
| fault | 0.2064 (10) | 0.4269 (4) | 0.2114 (8) | 0.4294 (3) | 0.4364 (2) | 0.1297 (12) | 0.2136 (7) | **0.4378 (1)** | 0.3836 (6) | 0.1458 (11) | 0.2114 (8) | 0.389 (5) |
| gas | 0.0265 (2) | 0.0017 (4) | 0.0016 (5) | 0.0016 (5) | 0.0018 (3) | **0.1382 (1)** | 0.001 (8) | 0.0009 (10) | 0.0009 (10) | 0.001 (8) | 0 (12) | 0.0016 (5) |
| glass | 0.1388 (2) | 0.0944 (8) | 0.068 (11) | 0.1033 (7) | 0.1268 (4) | 0.1318 (3) | **0.1393 (1)** | 0.0812 (10) | 0.1183 (6) | 0.1183 (6) | 0.0936 (9) | 0.1231 (5) |
| Glass2 | 0.1436 (7) | 0.2108 (2) | 0.2066 (3) | 0.1506 (5) | 0.1506 (5) | 0.0916 (11) | 0.0837 (12) | **0.2301 (1)** | 0.1115 (10) | 0.1301 (8) | 0.1301 (8) | 0.2034 (4) |
| HeartDisease | 0.4804 (9) | 0.5306 (5) | 0.5646 (1) | 0.5412 (3) | 0.5479 (2) | 0.0517 (12) | 0.5276 (6) | 0.5172 (8) | 0.4604 (11) | 0.4667 (10) | 0.5317 (4) | 0.5204 (7) |
| Hepatitis | 0 (11) | 0.2388 (8) | 0.3008 (2) | 0.2527 (5) | 0.2501 (6) | 0.2842 (3) | 0.259 (4) | 0.2407 (7) | 0.2012 (10) | 0 (11) | 0.2388 (8) | **0.329 (1)** |
| imgseg | 0.1062 (12) | 0.3506 (5) | 0.3635 (3) | 0.3485 (7) | 0.3498 (6) | **0.4699 (1)** | 0.2688 (8) | 0.3552 (4) | 0.1766 (11) | 0.1844 (9) | 0.1844 (9) | 0.3742 (2) |
| InternetAds | 0.2557 (10) | **0.5101 (1)** | 0.4136 (5) | 0.4431 (2) | 0.3385 (8) | 0.0097 (12) | 0.419 (4) | 0.3858 (6) | 0.3714 (7) | 0.2173 (11) | 0.3288 (9) | 0.4431 (2) |
| ionosphere | 0.7949 (4) | 0.81 (2) | 0.357 (10) | 0.7866 (5) | 0.3474 (11) | 0.2536 (12) | 0.81 (2) | 0.7858 (6) | 0.565 (9) | 0.6354 (8) | 0.6885 (7) | **0.8316 (1)** |
| landsat | 0.1561 (2) | 0.1291 (9) | 0.0941 (10) | 0.1325 (6) | 0.1314 (8) | **0.1628 (1)** | 0.1365 (3) | 0.1325 (6) | 0.0941 (10) | 0.1326 (5) | 0.0941 (10) | 0.1336 (4) |
| letter | 0.4889 (3) | 0.0866 (10) | 0.547 (2) | 0.0811 (11) | 0.201 (6) | **0.6958 (1)** | 0.1194 (8) | 0.0951 (9) | 0.0592 (12) | 0.3901 (4) | 0.1682 (7) | 0.3658 (5) |
| letter | **0.2038 (1)** | 0.0967 (10) | 0.1606 (2) | 0.0972 (8) | 0.0981 (7) | 0.0073 (11) | 0.1355 (4) | 0.0983 (6) | 0.1554 (3) | 0.0969 (9) | 0.0073 (11) | 0.1193 (5) |
| lympho | 0.7817 (8) | **1 (1)** | 0.8968 (7) | 0.9762 (4) | 0.6762 (10) | 0.2054 (12) | 0.9333 (6) | 0.9444 (5) | 0.753 (9) | **1 (1)** | 0.6471 (11) | **1 (1)** |
| magic | 0.1143 (8) | 0.1516 (4) | 0.1219 (7) | 0.1459 (5) | 0.1568 (2) | **0.2403 (1)** | 0.1104 (9) | 0.1352 (6) | 0.0866 (11) | 0.0479 (12) | 0.1096 (10) | 0.155 (3) |
| mammography | 0.0793 (11) | 0.2178 (2) | 0.1206 (10) | 0.1783 (4) | 0.1744 (6) | 0.0229 (12) | 0.1609 (8) | 0.1759 (5) | **0.3414 (1)** | 0.1537 (9) | 0.1692 (7) | 0.2033 (3) |
| mnist | 0.2211 (7) | 0.2435 (4) | 0.1589 (9) | 0.2421 (5) | 0.1096 (10) | 0.0982 (11) | 0.3418 (2) | 0.2635 (3) | 0.0785 (12) | 0.1787 (8) | 0.2333 (6) | **0.4136 (1)** |
| musk | 0.0662 (12) | 0.9147 (7) | 0.9994 (3) | 0.9964 (6) | 0.9996 (2) | 0.1654 (11) | 0.4462 (10) | 0.8162 (8) | 0.9994 (3) | **1 (1)** | 0.6592 (9) | 0.9992 (5) |
| opt.digits | 0.0764 (2) | 0.0673 (3) | 0.0609 (8) | 0.0664 (5) | 0.0664 (5) | 0.0607 (9) | 0.0673 (3) | 0.0588 (11) | 0.0554 (12) | 0.0592 (10) | 0.0619 (7) | **0.0822 (1)** |
| optdigits | 0.0321 (8) | 0.0449 (6) | 0.0222 (10) | **0.0639 (1)** | 0.0433 (7) | 0.0619 (2) | 0.0274 (9) | 0.0617 (3) | 0.0222 (10) | 0.0515 (5) | 0.053 (4) | 0.0219 (12) |
| pageb | 0.3763 (8) | 0.3615 (9) | 0.3615 (9) | 0.4594 (3) | 0.4377 (7) | 0.357 (10) | **0.4813 (1)** | 0.4787 (3) | 0.1635 (11) | 0.1092 (12) | **0.4813 (1)** | 0.458 (5) |
| PageBlocks | 0.2861 (9) | 0.495 (2) | 0.2654 (10) | 0.4596 (5) | 0.4767 (4) | 0.2333 (11) | **0.5039 (1)** | 0.4787 (3) | 0.2288 (12) | 0.4202 (7) | 0.3944 (8) | 0.4538 (6) |
| particle | 0.0633 (8) | 0.0724 (4) | 0.0564 (9) | 0.0841 (2) | 0.0668 (7) | **0.2732 (1)** | 0.0671 (6) | 0.0519 (10) | 0.0439 (11) | 0.0677 (5) | 0.0439 (11) | 0.0768 (3) |
| PenDigits | 0.0099 (5) | 0.0048 (7) | 0.094 (3) | 0.005 (6) | 0.0031 (9) | **0.5355 (1)** | 0.016 (4) | 0.0015 (10) | 0.0012 (11) | 0.0012 (11) | 0.1661 (2) | 0.0045 (8) |
| pendigits2 | 0.0528 (9) | 0.2013 (4) | 0.1143 (5) | **0.2843 (1)** | 0.0892 (6) | 0.0127 (12) | 0.0789 (7) | 0.263 (2) | 0.0414 (10) | 0.063 (8) | 0.0134 (11) | 0.2286 (3) |
| pima | 0.4239 (10) | 0.5139 (2) | 0.4044 (11) | 0.4674 (6) | 0.4674 (6) | 0.1632 (12) | 0.4504 (8) | 0.5073 (3) | **0.5586 (1)** | 0.4499 (9) | 0.4678 (5) | 0.505 (4) |
| Pima2 | 0.4606 (8) | 0.5175 (4) | 0.3471 (11) | 0.4674 (6) | 0.4467 (9) | 0.0511 (12) | 0.5314 (2) | 0.5097 (5) | 0.5254 (3) | 0.4446 (10) | 0.4699 (7) | **0.5434 (1)** |
| satellite | 0.3647 (11) | **0.6745 (1)** | 0.6131 (5) | 0.6315 (4) | 0.5931 (7) | 0.5675 (10) | 0.6082 (6) | 0.6697 (2) | 0.3571 (12) | 0.5757 (8) | 0.5729 (9) | 0.6515 (3) |
| satimage-2 | 0.0332 (10) | 0.9217 (3) | 0.7552 (8) | 0.8964 (5) | 0.9303 (1) | 0.0241 (11) | 0.8457 (7) | 0.9167 (4) | 0.0096 (12) | 0.8479 (6) | 0.6211 (9) | 0.9296 (2) |
| shuttle | 0.2052 (11) | 0.2249 (5) | 0.3363 (2) | 0.2382 (4) | 0.2232 (7) | **0.3736 (1)** | 0.2249 (5) | 0.2413 (3) | 0.1146 (12) | 0.2209 (9) | 0.2232 (7) | 0.2143 (10) |
| shuttle2 | 0.0981 (9) | 0.972 (3) | 0 (12) | **0.9724 (1)** | **0.9724 (1)** | 0.4726 (5) | 0.1199 (8) | 0.968 (4) | 0.3588 (6) | 0.0426 (10) | 0.1532 (7) | 0.0418 (11) |
| Shuttle3 | 0.3512 (2) | 0.0806 (6) | 0.2628 (4) | 0.0648 (8) | 0.0715 (7) | **0.5834 (1)** | 0.047 (11) | 0.0479 (10) | 0.0126 (12) | 0.1266 (5) | 0.0648 (8) | 0.3481 (3) |
| skin | 0.1102 (4) | 0.0972 (8) | **0.1538 (1)** | 0.1003 (7) | 0.1056 (5) | 0.066 (12) | 0.1156 (3) | 0.0761 (9) | **0.1538 (1)** | 0.076 (11) | 0.0761 (9) | 0.1007 (6) |
| smtp_n | 0.0012 (10) | 0.0046 (7) | 0.0035 (8) | 0.0074 (5) | 0.0087 (4) | **0.6709 (1)** | 0.0048 (6) | 0.0455 (3) | 0 (11) | 0 (11) | 0.0035 (8) | 0.2227 (2) |
| spambase | 0.012 (9) | 0.0238 (4) | 0.0036 (10) | 0.0228 (5) | 0.0248 (3) | 0.3843 (2) | 0.0176 (8) | 0.0189 (6) | 0.0036 (10) | 0.0036 (10) | **0.4243 (1)** | 0.0189 (6) |
| SpamBase2 | 0.3516 (10) | 0.4666 (7) | 0.3029 (11) | 0.4654 (8) | 0.4842 (6) | 0.5261 (4) | **0.5369 (1)** | 0.5283 (2) | 0.3015 (12) | 0.3802 (9) | 0.5283 (2) | 0.505 (5) |
| speech | 0.0284 (3) | 0.0193 (9) | 0.0324 (2) | 0.0246 (5) | 0.0147 (11) | 0.024 (6) | 0.0213 (8) | 0.0158 (10) | 0.0134 (12) | **0.0987 (1)** | 0.023 (7) | 0.0281 (4) |
| Stamps | 0.1453 (12) | 0.3326 (2) | 0.1569 (10) | 0.2957 (5) | 0.2856 (7) | 0.1532 (11) | **0.3383 (1)** | 0.2948 (6) | 0.2411 (9) | 0.3006 (3) | 0.2812 (8) | 0.2982 (4) |
| synthetic | **0.1552 (1)** | 0.1192 (5) | 0.1477 (2) | 0.1197 (4) | 0.113 (7) | 0.0938 (11) | 0.0986 (9) | 0.0974 (10) | 0.1032 (8) | 0 (12) | 0.1192 (5) | 0.1464 (3) |
| thyroid | 0.0356 (11) | 0.6295 (6) | **0.7751 (1)** | 0.6054 (7) | 0.3558 (8) | 0.069 (9) | 0.732 (3) | 0.697 (4) | 0.0265 (12) | **0.7751 (1)** | 0.0603 (9) | 0.6677 (5) |
| vertebral | 0.1135 (3) | 0.0917 (9) | 0.0918 (8) | 0.0925 (7) | 0.0996 (6) | **0.6885 (1)** | 0.0877 (11) | 0.0856 (12) | 0.0998 (5) | 0.1175 (2) | 0.1052 (4) | 0.0887 (10) |
| vowels | 0.3005 (4) | 0.1679 (5) | 0.3438 (3) | 0.0997 (8) | 0.0754 (9) | **0.8041 (1)** | 0.1031 (7) | 0.1452 (6) | 0.0193 (11) | 0.0279 (10) | 0.0193 (11) | 0.6355 (2) |
| wave | 0.0479 (3) | 0.0128 (9) | 0.01 (10) | 0.0115 (9) | 0.0124 (8) | **0.3835 (1)** | 0.0479 (3) | 0.0147 (6) | 0.0557 (2) | 0.0076 (12) | 0.01 (10) | 0.0339 (5) |
| Waveform | 0.0834 (4) | 0.0593 (8) | 0.1306 (3) | 0.0543 (9) | 0.0537 (10) | **0.4882 (1)** | 0.0478 (12) | 0.051 (11) | 0.2756 (2) | 0.0596 (7) | 0.0659 (5) | 0.064 (6) |
| wbc | 0.5394 (8) | 0.5783 (6) | **0.6497 (1)** | 0.594 (2) | 0.5226 (9) | 0.0503 (12) | 0.3401 (10) | 0.5736 (7) | 0.0989 (11) | 0.5853 (5) | 0.594 (2) | 0.594 (2) |
| WBC2 | 0.1038 (9) | **0.8643 (1)** | 0.0283 (10) | 0.8562 (2) | 0.7213 (3) | 0.2639 (8) | 0.0283 (10) | 0.3856 (7) | 0.0283 (10) | 0.6586 (5) | 0.6575 (6) | 0.6613 (4) |
| WDBC | **0.729 (1)** | 0.6773 (4) | 0.6699 (5) | 0.6449 (8) | 0.6962 (3) | 0.4283 (11) | 0.7252 (2) | 0.6579 (7) | 0.1268 (12) | 0.6202 (9) | 0.5957 (10) | 0.6622 (6) |
| Wilt | 0.1001 (2) | 0.0485 (7) | 0.0602 (5) | 0.0402 (11) | 0.0446 (9) | **0.6381 (1)** | 0.0653 (3) | 0.0485 (7) | 0.0446 (9) | 0.0644 (4) | 0.0495 (6) | 0.0402 (11) |
| wine | 0.0066 (12) | 0.0091 (8) | 0.0127 (3) | 0.0107 (4) | 0.0092 (7) | **0.644 (1)** | 0.0085 (9) | 0.0101 (6) | 0.0075 (10) | 0.0075 (10) | 0.2477 (2) | 0.0103 (5) |
| wine2 | 0.176 (8) | 0.2442 (4) | 0.1108 (9) | 0.2008 (5) | 0.3074 (3) | 0.3818 (2) | 0.0745 (10) | 0.1832 (6) | 0.0536 (12) | 0.0571 (11) | 0.1832 (6) | **0.8697 (1)** |
| WPBC | 0.2317 (9) | 0.2311 (10) | 0.2363 (5) | 0.223 (12) | 0.2487 (2) | **0.3333 (1)** | 0.2487 (2) | 0.2318 (8) | 0.2277 (11) | 0.2358 (6) | 0.2329 (7) | 0.2465 (4) |
| yearp | **0.4978 (1)** | 0.4922 (4) | 0.4937 (2) | 0.4921 (5) | 0.4805 (10) | 0.2076 (12) | 0.4877 (9) | 0.4901 (7) | 0.4758 (11) | 0.4905 (6) | 0.4937 (2) | 0.4881 (8) |
| yeast | 0.0596 (7) | 0.058 (9) | 0.0561 (12) | 0.0586 (8) | 0.064 (5) | **0.1227 (1)** | 0.0675 (2) | 0.0637 (6) | 0.0576 (10) | 0.0575 (11) | 0.0675 (2) | 0.0672 (4) |
| **Average** | 0.2154 (6.87) | 0.3197 (5.08) | 0.2689 (6.44) | 0.3137 (5.42) | 0.2892 (6.13) | 0.2704 (6.53) | 0.2814 (5.73) | 0.2981 (6.6) | 0.1946 (8.97) | 0.2594 (7.68) | 0.2582 (6.9) | **0.3382 (4.53)** |
| **STD** | 0.2027 | 0.2967 | 0.2602 | 0.2966 | 0.2712 | 0.221 | 0.2691 | 0.2839 | 0.2199 | 0.2813 | 0.2268 | 0.2882 |

Table 8: Pairwise statistical test results between METAOD and baselines by Wilcoxon signed rank test in ST. Statistically better method shown in **bold** (both marked **bold** if no significance). METAOD related pairs are surrounded by rectangles. METAOD achieves the highest MAP and best average rank, and statistically significantly better than all baselines except iForest.

| Method 1 | Method 2 | p-value | Method 1 | Method 2 | p-value | Method 1 | Method 2 | p-value |
|---|---|---|---|---|---|---|---|---|
| LOF (0.2154) | **IForest (0.3197)** | 0.0032 | **ME (0.2689)** | **ISAC (0.2892)** | 0.6817 | ISAC (0.2892) | **MetaOD (0.3382)** | 0.0006 |
| **LOF (0.2154)** | **ME (0.2689)** | 0.5258 | **ME (0.2689)** | **AS (0.2704)** | 0.9525 | **AS (0.2704)** | **SS (0.2814)** | 0.8471 |
| LOF (0.2154) | **GB (0.3137)** | 0.01 | **ME (0.2689)** | **SS (0.2814)** | 0.2345 | **AS (0.2704)** | **ALORS (0.2981)** | 0.5725 |
| **LOF (0.2154)** | **ISAC (0.2892)** | 0.1016 | **ME (0.2689)** | **ALORS (0.2981)** | 0.5031 | **AS (0.2704)** | **MetaOD_C (0.1946)** | 0.0667 |
| **LOF (0.2154)** | **AS (0.2704)** | 0.1683 | ME (0.2689) | **MetaOD_C (0.1946)** | 0.0026 | **AS (0.2704)** | **MetaOD_F (0.2594)** | 0.6112 |
| **LOF (0.2154)** | **SS (0.2814)** | 0.0575 | **ME (0.2689)** | **MetaOD_F (0.2594)** | 0.2389 | **AS (0.2704)** | **RS (0.2582)** | 0.8622 |
| LOF (0.2154) | **ALORS (0.2981)** | 0.0461 | **ME (0.2689)** | **RS (0.2582)** | 0.3672 | AS (0.2704) | **MetaOD (0.3382)** | 0.0009 |
| LOF (0.2154) | **MetaOD_C (0.1946)** | 0.0345 | ME (0.2689) | **MetaOD (0.3382)** | 0.0001 | **SS (0.2814)** | **ALORS (0.2981)** | 0.7604 |
| **LOF (0.2154)** | **MetaOD_F (0.2594)** | 0.6176 | **GB (0.3137)** | **ISAC (0.2892)** | 0.0849 | **SS (0.2814)** | MetaOD_C (0.1946) | 0.0001 |
| **LOF (0.2154)** | **RS (0.2582)** | 0.1735 | **GB (0.3137)** | **AS (0.2704)** | 0.4468 | **SS (0.2814)** | MetaOD_F (0.2594) | 0.0069 |
| LOF (0.2154) | **MetaOD (0.3382)** | 0.0001 | **GB (0.3137)** | **SS (0.2814)** | 0.5412 | **SS (0.2814)** | **RS (0.2582)** | 0.1419 |
| **IForest (0.3197)** | ME (0.2689) | 0.0484 | **GB (0.3137)** | **ALORS (0.2981)** | 0.1002 | SS (0.2814) | **MetaOD (0.3382)** | 0.0190 |
| **IForest (0.3197)** | GB (0.3137) | 0.047 | GB (0.3137) | **MetaOD_C (0.1946)** | 0.0001 | **ALORS (0.2981)** | MetaOD_C (0.1946) | 0.0001 |
| **IForest (0.3197)** | ISAC (0.2892) | 0.0127 | GB (0.3137) | **MetaOD_F (0.2594)** | 0.0001 | **ALORS (0.2981)** | MetaOD_F (0.2594) | 0.0280 |
| **IForest (0.3197)** | **AS (0.2704)** | 0.3714 | **GB (0.3137)** | **RS (0.2582)** | 0.0017 | **ALORS (0.2981)** | **RS (0.2582)** | 0.0524 |
| **IForest (0.3197)** | **SS (0.2814)** | 0.1942 | GB (0.3137) | **MetaOD (0.3382)** | 0.0030 | ALORS (0.2981) | **MetaOD (0.3382)** | 0.0001 |
| **IForest (0.3197)** | ALORS (0.2981) | 0.0012 | **ISAC (0.2892)** | **AS (0.2704)** | 0.7233 | MetaOD_C (0.1946) | **MetaOD_F (0.2594)** | 0.0342 |
| **IForest (0.3197)** | MetaOD_C (0.1946) | 0.0001 | **ISAC (0.2892)** | **SS (0.2814)** | 0.9513 | MetaOD_C (0.1946) | **RS (0.2582)** | 0.0115 |
| **IForest (0.3197)** | MetaOD_F (0.2594) | 0.0001 | **ISAC (0.2892)** | **ALORS (0.2981)** | 0.9357 | MetaOD_C (0.1946) | **MetaOD (0.3382)** | 0.0001 |
| **IForest (0.3197)** | RS (0.2582) | 0.0012 | **ISAC (0.2892)** | MetaOD_C (0.1946) | 0.0003 | **MetaOD_F (0.2594)** | **RS (0.2582)** | 0.4616 |
| **IForest (0.3197)** | **MetaOD (0.3382)** | 0.1129 | ISAC (0.2892) | **MetaOD_F (0.2594)** | 0.0159 | MetaOD_F (0.2594) | **MetaOD (0.3382)** | 0.0001 |
| **ME (0.2689)** | **GB (0.3137)** | 0.1002 | ISAC (0.2892) | **RS (0.2582)** | 0.0344 | RS (0.2582) | **MetaOD (0.3382)** | 0.0001 |