# OpenReview forum: "Automatic Unsupervised Outlier Model Selection"
_NeurIPS.cc/2021/Conference — NeurIPS 2021 Poster_

### Official Review · Reviewer_ZaAH · 2021-07-15

**Rating:** 6
**Confidence:** 4

**Summary:**

This paper proposes a data-driven method, METAOD to unsupervised outlier model selection problem based on meta-learning. METAOD makes use of many detection models on historical outlier detection benchmark datasets and selects an effective model to be employed on a new dataset without. Authors introduce meta-features that quantify the outlying characteristics of a dataset. Experiments have shown the promise of METAOD. The authors have implemented an open-source tool METAOD and our meta-learning database for practical use.


**Limitations And Societal Impact:**

See above.

**Main Review:**

The authors tackle a very challenging problem, unsupervised outlier model selection. Evaluating outlier detection methods is very challenging because of no ground truths and a lack of objective function makes unsupervised outlier model selection. The authors approach the problem as a collaborative filtering recommendation problem, and this is quite interesting.

Overall, I feel this method has some novelty. The selection of an outlier detection method is challenging, and the method proposed can be a practical solution. This paper utilises prior knowledge about outlier detection models (their performance on historical labelled datasets) and the similarity of an (unlabelled) test dataset to historical datasets to determine the best model for the test dataset. A key step is to find the similarity, i.e., meta-features. Two types of meta-features, statistical and landmark features, are designed to capture the similarities of two datasets in the general case and outlying aspects, respectively. The design of the method is reasonable.

However, I have some specific questions about the implementation.

The success of the proposed method depends on several factors, and the two major ones are as the following: 1) the outlying characteristics can be precisely quantified. 2) The performance of a method can be accurately measured.

The features and measures proposed are heuristics, and they need justification. The statistic features are very general and they may not reveal any outlying characteristics since otherwise, outlier detection can be based on these general statistics features. Furthermore, data sets have different dimensionalities. Can statistical properties of data sets with different dimensionalities be comparable?

Landmark features:  what are justifications for using the four OD methods? Do the features capture outlying chrematistics of data sets?

The meta-features can be many. Will this cause curse of dimensionality?

Equation (2) seems arbitrary for me. Why are the three functions used?

As the authors discussed, it is very difficult to quantify the performance of an outlier detection method. How do the authors resolve this problem?


**Time Spent Reviewing:**

3

---

> ### Author Response · Authors · 2021-08-10
> **Re: Official Review of Paper7338 by Reviewer ZaAH**
>
> **(1)** >> The features and measures proposed are heuristics, and they need justification. The statistic features are very general and they may not reveal any outlying characteristics since otherwise, outlier detection can be based on these general statistics features. Furthermore, data sets have different dimensionalities. Can statistical properties of data sets with different dimensionalities be comparable?
>
> We appreciate the clarification question on meta-feature extraction. The statistical features used in MetaOD (Table 3 in Appendix) are chosen as they are the most commonly used ones in meta-learning literature [1,2,3], with proven effectiveness in various scenarios. These features capture basic distributional properties, such as the dispersion and variability of the data, which are closely related to the inlier and outlier distributions within a dataset. We agree that generic statistical meta-features may be insufficient to depict the outlying characteristics, and therefore build specialized landmarker features to enrich statistical meta-features in MetaOD.
>
> By combining the statistical meta-features with the specialized landmark features, the meta-features in MetaOD can find dis/similar datasets with varying numbers of samples and dimensionalities. Fig. 1 (left) in the paper shows that we can successfully identify clusters of similar datasets in the POC using the meta-features. Notably, meta-features work for these POC datasets with varying numbers of dimensionalities, ranging from 4 to 404.
>
> **(2)** >> Landmark features: what are justifications for using the four OD methods? Do the features capture outlying characteristics of data sets?
>
> Thanks for this important question. As we discussed in Section 3.3 and other meta-learning literature [1,3] have found, the optimal set of meta-features is application-dependent. This is also true for landmarker features. As the first work that builds landmarker features for outlier detection tasks (to our best knowledge), we choose the four OD algorithms due to their efficiency and diversity (as a group). First, they are all fast algorithms and able to handle large, high-dimensional datasets. This makes the meta-feature generation efficient and practical in the real world, leading to less than 1 second model selection. Second, these four OD algorithms as a group show decent diversity to capture rich outlying characteristics: (1) iForest is a tree-based subspace ensemble method, (2) PCA is a deterministic linear method; (3) HBOS treats each dimension independently and (4) LODA is a lightweight histogram-based ensemble method. Again, the visualization in Fig. 1 (left) illustrates that using the current meta-features can identify correct clusters of “sibling datasets” in the POC testbed.
>
> We also agree that an enhanced understanding of meta-features is important in model selection research. Along with existing literature [1,4], we will add this open direction to future work.
>
> **(3)** >> The meta-features can be many. Will this cause curse of dimensionality?
>
> Thanks for the question on the potential curse of dimensionality of meta-features. Indeed, if one builds thousands of meta-features, the curse of dimensionality may cause problems. To our best knowledge, most meta-learning research in this line builds no more than a few hundred meta-features. In MetaOD, we use only 200 meta-features to represent a dataset.
>
> More importantly, we do not use raw meta-features directly for measuring dataset similarity, which may be susceptible to the curse of dimensionality. In MetaOD, we map the meta-features to the corresponding dataset vector $\mathbf{U_{test}}$ first through a random forest regressor, which has built-in feature selection and is capable of handling a reasonable number (several hundreds) of meta-features.
>
> **(4)** >> Equation (2) seems arbitrary for me. Why are the three functions used?
>
> We will make Eq. (2) clearer in the revision. Basically, it describes the sequence of steps that MetaOD takes for selecting the model for a test data $\mathbf{X_{test}}$ (see Section 3.2.2 for details). First, it computes the corresponding meta-features as $\mathbf{M_{test}} := \psi (\mathbf{X_{test}})$  where $\psi$ denotes the meta-feature extraction process. Then, the corresponding dataset vector $\mathbf{U_{test}}$  is generated by the successive application of the embedding operator ($\phi(\cdot)$) and then the regressor ($f(\cdot)$). That is, $f(\phi(\psi(\mathbf{X_{test}})))$ in Eq. (2) refers to the generation process of $\mathbf{U_{test}} := f(\phi(\mathbf{M_{test}}))$ . We could then get the predicted model set performances via a dot product,  $\mathbf{P_{test}} := \mathbf{U_{test}} \mathbf{V}^T$. Finally, argmax in Eq. (2) indicates that we select the model $j$ from the pool that has the highest predicted model performance for $\mathbf{X_{test}}$.
>
> **(5)** >> As the authors discussed, it is very difficult to quantify the performance of an outlier detection method. How do the authors resolve this problem?
>
> In experiments, we perform leave-one-dataset-out based evaluation. That is, we treat one dataset at a time as the test/newcoming dataset, and the remaining datasets as (meta) train datasets. The ground-truth labels of the test dataset are hidden from MetaOD during model selection. We use those only to evaluate the performance of the *selected* model by MetaOD in order to report the quality/performance of models as selected by MetaOD. This is the standard practice in unsupervised outlier detection -- ground-truth labels are only used for evaluation and reporting purposes, and _not_ during the main detection (or in our case, selection) task.
>
>
> [1] Vanschoren, J., 2018. Meta-learning: A survey. arXiv preprint arXiv:1810.03548.
>
> [2] Rivolli, A., Garcia, L.P., Soares, C., Vanschoren, J. and de Carvalho, A.C., 2018. Towards reproducible empirical research in meta-learning. arXiv preprint arXiv:1808.10406, pp.32-52.
>
> [3] Hutter, F., Kotthoff, L. and Vanschoren, J., 2019. Automated machine learning: methods, systems, challenges (p. 219). Springer Nature.
>
> [4] Alcobaça, E., Mantovani, R.G., Rossi, A.L. and De Carvalho, A.C., 2018, October. Dimensionality reduction for the algorithm recommendation problem. In *2018 7th Brazilian Conference on Intelligent Systems* (pp. 318-323). IEEE.

---

### Official Review · Reviewer_EPJc · 2021-07-16

**Rating:** 7
**Confidence:** 4

**Summary:**

The paper proposes a method, named MetaOD, which for each input data set selects the most suited outlier detection model together with its relevant hyper-parameter values from a set of given models. The selection is done in a supervised manner. MetaOD works as follows. During the meta-training stage, it considers a set of $m$ models $M = \{M_1, ..., M_m\}$ and a set of $n$ data sets $D = \{D_1, ..., D_n\}$. Each model $M_i$ includes model architecture and model configuration (hyper-parameter values). MetaOD computes performance matrix $P$ where cell $P_{ij}$ is the performance of method $M_j$ on data set $D_i$. Then it learns a data matrix $U \in R^{n \times k}$ and a model matrix $V \in R^{m \times k}$ where $U_i \times V_j^T$ approximates $P_{ij}$. The functions learnt from $U$ and the matrix $V$ are then used during test time to choose the best model (together with its configuration) for each input data set.

**Ethical Concerns:**

As mentioned above.

**Limitations And Societal Impact:**

The authors do address the technical limitations of the paper that they are able to identify.

The authors do not identify any societal impact of the paper. However, one thing they could explore, perhaps in future work, is how biases embedded in base OD models and historical data sets could influence the selection of best model for incoming test sets.

**Main Review:**

Unsupervised outlier detection is a challenging task as it lacks a universal objective function and reliable evaluation data sets. Methods proposed so far have their own notion of outlier scores and do not consistently have best performance. This makes it hard in practice to choose a suitable OD method for the data set at hand. As a plus point, the paper tackles the issue by mapping unsupervised OD model selection to the collaborative filtering cold-start problem, enabling the transfer of well-established solutions in CF to OD.

I find the solution of finding $U$ and $V$ through matrix factorisation on $P$ (solved by alternating optimisation) straightforward and standard. I have two concerns though. First, while $V$ is used directly during test time, $U_{\mathit{test}}$ is just a mapping of the test set through a series of three functions, one of which is regression function $f(\cdot)$. Function $f(\cdot)$ on the other hand is not optimised together with $U$ and $V$ during meta-training stage, and hence might be sub-optimal to reconstructing matrix $P$. Is there a way to learn $f(\cdot)$ during alternating optimisation? Second, MetaOD focuses on selecting the best model. It would be more interesting if the authors solve the more general problem of selecting top-k models (to build an ensemble) and make the top-1 a special case.

The experiments look extensive with 2 settings: POC and stress tests. I find the baselines and data sets sufficient. In addition, MetaOD achieves good empirical performance and takes about 1 second to choose the best model for each input data set.

The writing has room for improvement. For instance, a table summarising the notations used would make the paper more readable. The part about data-specific meta-feature extraction is important and should be explained in the main part of the paper (e.g. through an example data set).

**Time Spent Reviewing:**

5

---

> ### Author Response · Authors · 2021-08-10
> **Re: Official Review of Paper7338 by Reviewer EPJc**
>
> **(1)** >> First, while $\mathbf{V}$ is used directly during test time,  $\mathbf{U_{test}}$ is just a mapping of the test set through a series of three functions, one of which is regression function $f(\cdot)$ . Function $f(\cdot)$ on the other hand is not optimised together with $\mathbf{U}$ and $\mathbf{V}$ during meta-training stage, and hence might be sub-optimal to reconstructing matrix $\mathbf{P}$. **Is there a way to learn $f(\cdot)$ during alternating optimisation**?
>
> Thanks for pointing out this interesting direction! It is intriguing to see the joint optimization idea by combining the loss from both matrix factorization and regressor training. In MetaOD, the mapping function $f(\cdot)$ is a flexible choice and can be any multi-output regression model, where the training process is to minimize its loss function regarding the input (meta-features of the meta-train datasets) and output (dataset matrix $\mathbf{U}$). Without committing to a specific regression model $f(\cdot)$ and a corresponding loss function, the joint optimization will be less obvious. Notably, the regression model used in the experiments (being a high-capacity ensemble, i.e. random forests) already yields an excellent fit, as such, the space for improvement for joint optimization may be limited. We plan to list the reviewer’s joint optimization idea in the future directions for exploration.
>
> **(2)** >> Second, MetaOD focuses on selecting the best model. It would be more interesting if the authors solve the **more general problem of selecting top-k models** (to build an ensemble) and make the top-1 a special case.
>
> This is a great suggestion! We agree that MetaOD can be extended to top-k model selection followed by an ensemble, leading to a new line of research. MetaOD provides the predicted performance of all models on the test dataset, after which one may choose various ways to do the aggregation on top k models, e.g., simple averaging or more complex second-round selection methods. We will add this in our future work.
>
> Having said that, note that top-k model selection would incur the additional challenges of deciding (i) the value for k as well as (ii) how to combine/aggregate the k selected models, whereas in its current form MetaOD is parameter-free --which we think is  a desirable property for an automation method itself.
>
> **Clarity Improvement**
>
> **(1)** >> The writing has room for improvement. For instance, a table summarising the notations used would make the paper more readable. The part about data-specific meta-feature extraction is important and should be explained in the main part of the paper (e.g. through an example data set).
>
> We plan to move Figure 5, currently in Appendix, to the main paper in revision, to depict the main flow of MetaOD along with annotating key notation on the figure. To make the meta-feature extraction clearer, we will extend Section 3.3 with more details, and an example if space allows.
>
> **Limitations & Societal Impact**
>
> **(1)** >> The authors do not identify any societal impact of the paper. However, one thing they could explore, perhaps in future work, is how biases embedded in base OD models and historical data sets could influence the selection of best model for incoming test sets.
>
> Outlier detection is a critical problem for many domains, including finance, security, medicine, etc. Hence effectively detecting anomalies has direct and indirect implications for society. Our work aims to enable better detection performance, through being able to systematically select a better model to employ on a given/new task. When deployed and successful, therefore, we expect MetaOD to help “lift up” detection quality, which could translate to saved money (e.g. credit card fraud), better security (e.g. surveillance), better healthcare (e.g. patient monitoring at the ICU), etc. depending on the many applications of outlier detection.
>
> We appreciate the suggestion on exploring the potential bias and fairness issues in OD model selection. We will list this as the future direction, along with the related literature [1,2].
>
> [1] Davidson, I. and Ravi, S.S., 2020. A framework for determining the fairness of outlier detection. In *Proceedings of the 24th European Conference on Artificial Intelligence (ECAI2020)* (Vol. 2029).
>
> [2] Shekhar, S., Shah, N. and Akoglu, L., 2021. FAIROD: Fairness-aware Outlier Detection. *AAAI/ACM Conference on AI, Ethics, and Society (AIES)*.

---

### Official Review · Reviewer_4gCH · 2021-07-19

**Rating:** 7
**Confidence:** 5

**Summary:**

The paper proposes a model selection approach called METAOD for outlier detection. METAOD maintains historical performances of various outlier detection models on different datasets, where the performance of a detection model on a new dataset is estimated based on the historical information. The setting is thus similar to meta-learning. METAOD is motivated by Collaborative Filtering, where the compatibility of a user (i.e., a dataset) and an item (i.e., a model) is estimated. A set of meta-features are crafted for charactering a dataset. Experiments show that METAOD benefits outlier detection model selection, compared with no model-selection baselines.

**Limitations And Societal Impact:**

Yes

**Main Review:**

Originality: The paper targets a novel task, i.e., model selection for outlier detection (unsupervised). The task poses a new challenge that traditional model selection methods could not tackle, that it is hard to evaluate models on datasets without ground truths.

Quality: The proposed technique is, in general, intuitive and reasonable. A CF-based matrix factorization is used to learn the latent space for measuring model-data similarity. A forward mapping function f is learned to transform data features to the latent space. A PCA step is included as pre-processing. In the CF step, a back-propagatable DCG-based loss is developed to replace traditional MSE loss. However, it would be better to report this advantage through experiments.

Clarity: The paper is well-structured and well-written.

Significance: Model selection is a crucial task for outlier detection. The paper (partially) tackles the important problem, as long as practitioners have already accumulated a significant number of historical datasets and evaluated models. The testbed setup consists of two scenarios (POC and ST), which is an interesting setting.

**Time Spent Reviewing:**

3

---

> ### Author Response · Authors · 2021-08-10
> **Re: Official Review of Paper7338 by Reviewer 4gCH**
>
> **(1)** >> In the CF step, a back-propagatable DCG-based loss is developed to replace traditional MSE loss. However, it would be better to report this advantage through experiments.
>
> We appreciate the clarification request on the advantages of using DCG-based loss over MSE loss. We conducted additional experiments for it. By controlling all other conditions, MetaOD (with the rank-based loss) outperforms the MetaOD variant with the MSE loss. In ST, the former achieves a higher mean AP (higher is better) than the latter (0.3382 vs. 0.3012). The pairwise Wilcoxon signed rank test also shows that the difference is statistically significant (p=0.0001).
>
> **(2)** >> The paper (partially) tackles the important problem, as long as practitioners have already accumulated a significant number of historical datasets and evaluated models.
>
> The reviewer is right -- MetaOD will excel more, the larger the meta-train database (i.e. prior experience) is -- which is the key driving factor behind the prowess of meta-learning. Exactly for this reason, we have made MetaOD an open platform for the community to contribute: new datasets, new models, etc. so as to accumulate all of the community’s effort in an organized way for this challenging yet important problem.

---

### Official Review · Reviewer_21Jt · 2021-07-28

**Rating:** 6
**Confidence:** 3

**Summary:**

This paper presents a data-driven approach to model selection for unsupervised outlier detection. The proposed method, METAOD takes a large number of training datasets with outlier labels and a pool of outlier detection models (methods plus their parameters) and uses a matrix factorization method to learn a model performance predictor, which, for given a test dataset, can predict/select a high performing model from the model pool without requiring test time model evaluation.

The performance of METAOD is evaluated with two implementations with the same pool of 300+ models (8 distinct outlier detection methods with different parameter configurations) but two different groups of training datasets.

**Limitations And Societal Impact:**

The authors should provided more discussions on the limitations of the proposed method, especially the limitations related to the applicability of the method.

Currently there are no discussions provided on the societal impact of the paper.

**Main Review:**

Please find below some detailed comments on the originality, significance, quality and clarity of the paper.

1. Originality and significance

Strong points:

(a) The idea of using meta-learning for selecting outlier detection models is interesting and sounds novel to me.

(b) There is not a model that can suit all outlier detection problems, and it is important and a good idea to try to select a high performing model for each specific dataset.

(c)  It is infeasible to conduct test time model selection or evaluation for outlier detection problems, hence using historical data and the similarity between test data and historical data is a good direction to explore.

Weakness:

(a) For the proposed method/framework to be applicable, it would be good if the authors could discuss more about the limitations of the proposed method, in particular, it would be ideal if some guidelines/analysis or evaluation can be provided regarding the extend of similarity of test and train data for the proposed method to be effective.

(b) In Section 2.2, more discussions should be given on the difference between the key ideas/methods used by METAOD and the existing methods which use CF solutions for ML model selection.

2. Quality

Strong points:

(a) The overall idea of using meta learning to capture models' experience on outlier detection in different historical data.

(b) The linking between unsupervised outlier detection model selection and collaborative filtering.

(c) The creation and use of meta-features

(d) A comprehensive open-source implementation and evaluation

Weakness:

(a) The success of METAOD relies on the similarity between test data and training data. Given the "unknown" nature of outliers, it is quitely likely that in practice such similarity may not exist or may be very weak. As mentioned above, discussions and ideally evaluations on this applicability problem of METAOD should be provided, and the authors may also want to consider extend METAOD with the ability to output "null" when no good models can be picked up from the model pool.

(b) How the different factors, e.g. ways of discretizing hyperparameters, number of models to include in the pool, number of training datasets, performance evaluation metrics used for creating P, etc. will affect the overall performance of METAOD?

3. Clarity

Overall the paper is not difficult to follow, but the presentation can be improved by providing more clarity on some of the details and correcting the English problems, as detailed below.

(a) The description in Section 3.2 sometimes is not easy to follow. It would be better if an outline of the process and key parts of METAOD could be given at the start of the section. I understand that Fig. 5 in Appendix D may use lots of space, but it helps me a lot to understand the working of METAOD. The authors may consider to have a simplified version of Figure 5. The current writing of Section 3.2 is a bit loose and it could be tighten up too  to leave room to move Figure 5 to the main text and organise the writing in this section around the Figure.

(b) There are some typos and other English problems. Examples are:

*Line 11: selecting -> select
*Line 27: winner -> winners
*Line 179: largest performance -> best performance?
*Line 282: affect -> effect
*Line 286: footnotes 6,7 and 8 are not included in the main paper
*Line 309: whom -> who

**Time Spent Reviewing:**

5

---

> ### Author Response · Authors · 2021-08-10
> **Re: Official Review of Paper7338 by Reviewer 21Jt**
>
> **Originality**:
>
> **1(a)** >> For the proposed method/framework to be applicable, it would be good if the authors could discuss more about the limitations of the proposed method, in particular, it would be ideal if some **guidelines/analysis or evaluation can be provided regarding the extend of similarity of test and train data for the proposed method to be effective**.
>
> This is a great point! The effectiveness of meta-learning depends on the similarity between the test dataset and the meta-train database, and that is why we create two testbeds to better understand it. The POC testbed results show that 4 similar datasets could already yield excellent results, comparable to selecting the 4th best model from the pool of 302 models (top 1%). The ST testbed is closer to the real-world setting with the datasets from (three different) independent sources. In this challenging testbed, we have limited knowledge of the similarity between the test datasets and the meta-train database. Even then, MetaOD can still select the top 20% models consistently in ST. The results from both testbeds show that MetaOD is a reasonable choice with limited knowledge of dataset similarity between the test dataset and the meta-train database.
>
>
> **1(b)** >> In Section 2.2, more discussions should be given on the difference between the key ideas/methods used by METAOD and the existing methods which use CF solutions for ML model selection.
>
> As we have presented in Sections 3.3 and 3.4, the major differences include: (1) MetaOD builds specialized landmarker features tailored for capturing outlying characteristics of a dataset, while the existing ML model selection mainly uses generic statistical features and (2) MetaOD uses a customized (back-propagatable/smooth) rank-based loss in CF for more effective top-1 optimization, while existing approaches mainly leverage mean squared loss (MSE). By controlling all other conditions, MetaOD (with the rank-based loss) outperforms the MetaOD variant with the MSE loss In ST. The former comes with a higher mean AP (higher is better) than the latter (0.3382 vs. 0.3012). The pairwise Wilcoxon signed rank test also shows that the difference is statistically significant (p=0.0001). These two improvements together enable MetaOD to outperform the baselines in OD model selection.
>
> Another key difference is that in traditional ML model selection scenarios, with or without CF,  access to some validation data _with_ labels is assumed and leveraged in various ways. This opens the door for evaluation of some candidate models during selection. In contrast, for MetaOD there exists no labels for newcoming tasks and no models can be evaluated or compared to one another directly for model selection. As such, unsupervised outlier model selection presents a strictly harder problem.
>
> **Quality**
>
> **2(a)** >> The success of METAOD relies on the similarity between test data and training data. Given the "unknown" nature of outliers, it is quietly likely that in practice such similarity may not exist or may be very weak. As mentioned above, discussions and ideally evaluations on this applicability problem of METAOD should be provided, and the authors may also want to consider extend METAOD with the ability to output "null" when no good models can be picked up from the model pool.
>
> The prowess of meta-learning to work for our problem is the reliance on similar past experience. It is exactly why we set up 2 testbeds for evaluation to study this phenomenon: a controlled testbed with highly similar “sibling” datasets (POC) and a “wild” testbed that combined all benchmark datasets (that had ground-truth labels) from 3 different dataset repositories (ST). Our evaluation reveals that MetaOD excels when train/test dataset similarity is high, as in POC. Moreover, even in ST, where dataset similarity is weaker (recall Figure 1), MetaOD significantly outperforms the baselines.
>
> Returning “Null”: This is a great suggestion! One way to approach this would be looking at the similarity distribution of meta-train datasets and checking the value of a test dataset against that distribution to decide if the value is likely to come from the same distribution, or otherwise returning “null”. More sophisticated ideas from “conformal prediction” can also be explored, toward estimating precise levels of confidence in new predictions. This certainly opens a practically useful and interesting direction to pursue in (our) future work.
>
> **2(b)** >> How the different factors, e.g. ways of discretizing hyperparameters, number of models to include in the pool, number of training datasets, performance evaluation metrics used for creating P, etc. will affect the overall performance of METAOD?
>
> Currently, MetaOD can select among a finite-size pool of models with pre-set hyperparameters. A more granular discretization of the hyperparameter space, and inclusion of more detector families, may enable identification of better models at the expense of increased pool/candidate size -- which in turn would likely make the selection problem relatively more challenging. We have not particularly studied how MetaOD performance would change w.r.t. pool size/number of meta-train datasets/etc., however, we tried to set up a large pool of 300+ models with 8 different detector families to make the selection task non-trivial, and constructed our meta-train database from all publicly-available benchmark datasets from 3 different repositories.
>
> In general, we believe a larger meta-train database could help push the effectiveness of MetaOD further, without affecting the computational effort at test/selection time much.  That is exactly why we built MetaOD as an open platform for the community to contribute more datasets and/or models.
>
>
> **Clarity Improvement**
>
> **3(a)** >> The description in Section 3.2 sometimes is not easy to follow. It would be better if an outline of the process and key parts of METAOD could be given at the start of the section. I understand that Fig. 5 in Appendix D may use lots of space, but it helps me a lot to understand the working of METAOD. The authors may consider to have a simplified version of Figure 5. The current writing of Section 3.2 is a bit loose and it could be tighten up too to leave room to move Figure 5 to the main text and organise the writing in this section around the Figure.
>
> We thank the reviewer for the ideas of providing key parts/overview upfront and then organizing the writing in Sec. 3.2 around Figure 5 (currently in Appendix). We will revise the final version of our paper exactly based on this suggestion.
>
> **3(b)** >> There are some typos and other English problems.
>
> Thanks, we will do another round of proofreading to fix all the grammatical issues.
>
>
> **Limitations & Societal Impact**
>
> **(1)** >> The authors should provide more discussions on the limitations of the proposed method, especially the limitations related to the applicability of the method.
>
> In the paper, we mention one limitation of MetaOD as working with a finite pool of models with pre-specified hyperparameters, even though the hyperparameter (HP) space is infinite in theory. Most supervised ML model selection works address the continuous HP selection problem, using ideas from Bayesian optimization that necessitate model evaluations at carefully-selected HPs, which is not possible for MetaOD.  Having said that, we tried to make MetaOD practically relevant by using 8 SOTA detector families and a generous discretization of respective hyperparameters.
>
> Another limitation is outlined by the reviewer: Currently, MetaOD would return an answer (a selected model) no matter how similar a test dataset is to meta-train datasets. We aim to tackle this problem systematically in our ongoing follow-up work, by building a “self-aware”* selection method that can opt out of unconfident predictions by saying “I don’t know” [1].
>
>
> **(2)** >> Currently there are no discussions provided on the societal impact of the paper.
>
> Outlier detection is a critical problem for many domains, including finance, security, medicine, etc. Therefore, effectively detecting anomalies has direct and indirect implications for society. Our work aims to enable better detection performance, through being able to systematically select a better model to use for a given/new task. When deployed and successful, therefore, we expect MetaOD to help “lift up” detection quality, which could translate to saved money (e.g. credit card fraud), better security (e.g. surveillance), better healthcare (e.g. patient monitoring at the ICU), etc. depending on the many applications of outlier detection.
>
> [1] Li, L., Littman, M. L., Walsh, T. J., & Strehl, A. L. (2011). Knows what it knows: a framework for self-aware learning. *Machine learning*, 82(3), 399-443.

---

> > ### Comment · Reviewer_21Jt · 2021-08-23
> > **Thanks for your response**
> >
> > Thanks for the detailed and helpful response. I believe the work has its novelty and practical value, but at the same time, there are rooms for improvement. So I'd keep my score unchanged.

---

### Author Response · Authors · 2021-08-10
**Summary of Our Responses and Contributions**

We sincerely thank all the reviewers for encouraging and insightful comments. We have carefully read through them and provide corresponding responses individually.

We also want to reiterate the major contributions of our paper as follows:

- **Novelty**: We proposed MetaOD, (to our knowledge) the first effort on unsupervised model selection for outlier detection (OD) tasks. *As Reviewers ZaAH and EPJc affirm, this is a “very challenging problem/task” and we appreciate that Reviewers 21Jt, 4gCH, and ZaAH consider the idea of using meta-learning to tackle unsupervised outlier model selection as “a novel approach”*.
- **Problem Formulation**: We establish a correspondence between unsupervised outlier model selection (UOMS) and collaborative filtering under cold-start, where the new task “better likes” a model that performs better on similar historical tasks. *Reviewers ZaAH, 4gCH, and EPJc also think our approach to the problem is “quite interesting”, “intuitive”, “enabling the transfer of well-established solutions”*.
- **Specialized Meta-features**: We design novel meta-features to capture the outlying characteristics in a dataset toward effectively quantifying task similarity specifically among OD tasks. *We are glad Reviewer 21Jt thinks that “the creation and use of meta-features” as one of the strong points of the paper*.
- **Effectiveness and Efficiency**: Through extensive experiments on two benchmark testbeds, we show that selecting a model by MetaOD for each given task significantly outperforms competitive baselines. We appreciate that Reviewer EPJc finds that “experiments look extensive with 2 settings” and “the baselines and data sets sufficient”. *As Reviewer EPJc has outlined, “MetaOD achieves good empirical performance [**effective**] and takes about 1 second [**efficient**]  to choose the best model for each input data set.”*

---

### Decision · Program_Chairs · 2021-09-27

**Decision:**

Accept (Poster)

**Comment:**

All reviewers have praised the importance of selecting outlier detectors in an automatic way and recognized the proposed meta-learning solution as a novel contribution.

Important points were raised during the reviewing and discussion period. They concern:
  - the influence of the distribution similarity between train and test to make the meta-learning succeed,
  - the effect of some hyperparameters involved and
  - some rewriting of the presentation to ease reading (notation + English grammar typos) and to better motivate the use of a collaborative filtering approach.

The paper is accepted subject to the above points are addressed in the camera-ready.